# From pixels to connections: exploring in vitro neuron reconstruction software for network graph generation
Cassandra Hoffmann [1] ✉, Ellie Cho[2], Andrew Zalesky [1,3] & Maria A. Di Biase[1,4,5]

Digital reconstruction has been instrumental in deciphering how in vitro neuron architecture shapes information flow. Emerging approaches reconstruct neural systems as networks with the aim of understanding their organization through graph theory. Computational tools dedicated to this objective build models of nodes and edges based on key cellular features such as somata, axons, and dendrites. Fully automatic implementations of these tools are readily available, but they may also be purpose-built from specialized algorithms in the form of multi-step pipelines. Here we review software tools informing the construction of network models, spanning from noise reduction and segmentation to full network reconstruction. The scope and core specifications of each tool are explicitly defined to assist bench scientists in selecting the most suitable option for their microscopy dataset. Existing tools provide a foundation for complete network reconstruction, however more progress is needed in establishing morphological bases for directed/weighted connectivity and in software validation.

Two-dimensional (2D) neuron cultures offer a precise, efficient, and cost-effective model in modern neuroscience. In this context, microscopy images function as quantitative datasets harnessed for analysis through a growing body of neuroinformatic tools (Fig. 1a). Most metrics deducible from neuron reconstructions characterize cell morphology, such as neurite area or length, as indicators of cell development and health[1–3]. However, these discrete attributes are often agnostic to their broader biological context. For example, healthy neurite pruning and fasciculation often translate to reduced neurite area and length. Therefore, an incorrect impression of neuron viability is easily conceivable if these metrics are solely relied upon for analysis.

As a complimentary approach, in vitro neuron ensembles may be interpreted as systems whose topology is organized to optimally facilitate function. This concept is operationalized through network graphs. In mathematical terms, networks are graph theoretical objects comprised of $N$ units as nodes and $N{\times}N$ internodal relationships as edges (Fig. 1b). Edges may be weighted, where a value denotes their relative strength (Fig. 1c) and/ or directed, where they possess orientation (Fig. 1d). Of primary appeal to network analysis is the ability to reveal patterns of energy and information transfer that underpin overall system performance. This is not domain-specific, and in fact seminal work lay outside the field of neuroscience.

In 1998, Watts and Strogatz[4] highlighted the ubiquity of so-called small world topology in systems that optimise data propagation – such as social networks, food chains, and electronic power grids – by balancing long-distance signalling with specialized local clique signalling[4].

Applications of network science to macroscale brain systems have proven fruitful in characterizing the structural and functional topologies of different states, with a focus on psychiatric illnesses such as major depressive disorder[5,6], schizophrenia[7], and obsessive compulsive disorder[8], among others[9–11]. Recently these applications have been extended to microscale neural systems to probe the molecular mechanisms underlying wider brain structure and function. For example, in vitro neuron network studies have documented the spontaneous emergence of organized electrophysiological activity in culture[12–17], which is shaped by electrical[18,19] and chemical perturbation[20] in a way that informs our understanding of in vivo dynamics. However, despite multimodal studies suggesting a substantive role of physical connectivity in functional networks[21,22], only a limited selection of studies have characterized anatomical neuron network structure. Of these few, notable work by De Santos-Sierra et al. conducted between 2014 and 2019 documented the self-organization of locust neurons over 18 days of maturation[23–25]. An increase in small world-related graph metrics was observed, indicating a shift towards modular cell organization and efficient

[1]Systems Neuroscience Lab, Melbourne Neuropsychiatry Centre, Department of Psychiatry, The University of Melbourne, Parkville, Australia. [2]Biological Optical Microscopy Platform, University of Melbourne, Parkville, Australia. [3]Department of Biomedical Engineering, The University of Melbourne, Parkville, Australia. [4]Stem Cell Disease Modelling Lab, Department of Anatomy and Physiology, The University of Melbourne, Parkville, Australia. [5]Psychiatry Neuroimaging Laboratory, Department of Psychiatry, Brigham and Women's Hospital, Harvard Medical School, Boston, USA. ✉e-mail: crho@student.unimelb.edu.au

**Fig. 1 | Neuron connectivity represented through network graphs. a** A confocal microscopy image of stem cell-derived neurons cultured in monolayer format. Neurons immunostained for neuronal marker β-Tubulin III (green) and nuclear dye Hoechst 33342 (blue) exhibit self-organization. **b** Schematic graph representing neuronal connectivity, comprised of nodes (illustrated in green) and edges (illustrated in orange). An unweighted, undirected graph serves to represent basic relationships between neuronal elements as nodes. **c** A weighted graph incorporates edge values to confer the strength of internodal relationships. **d** A directed network incorporates edge orientation to confer the direction of internodal relationships.

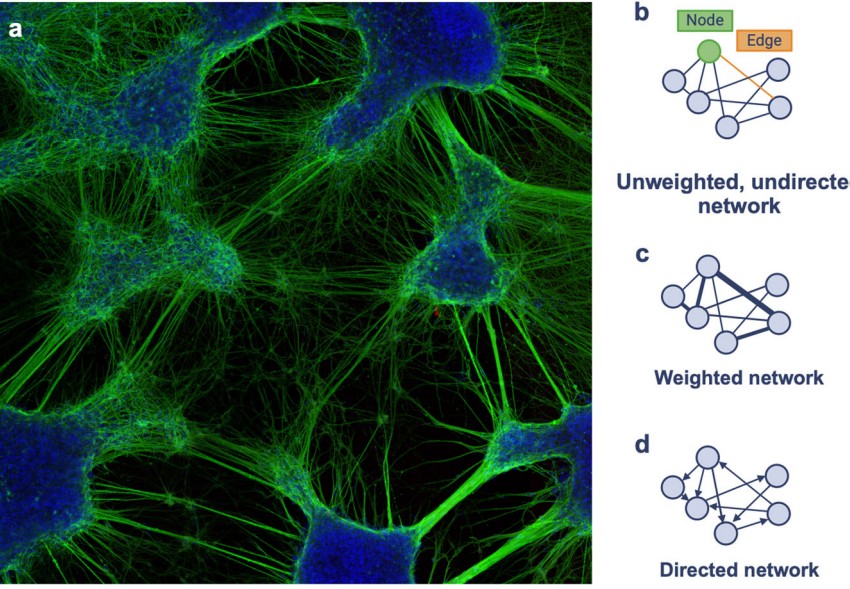

**Fig. 2 | A schematic pipeline for network reconstruction. a** Workflows begin with the acquisition of neuron images through microscopy. **b** Pre-processing techniques aim to improve image signal-to-noise ratio and reduce ambiguities. **c** Segmentation creates a mask of neuron morphology (top left of panel), which can be skeletonized (bottom left of panel). Tracing creates a tree of neuron centrelines (right half of panel). **d** Morphological labelling resolves the neuron mask into features such as somata (orange) and neurites (yellow). **e** Post-processing methods refine or extract features, such as branch points (blue) from the neuron skeleton (yellow). **f** Network reconstruction creates a model representing morphological features as nodes and their relationships as edges.

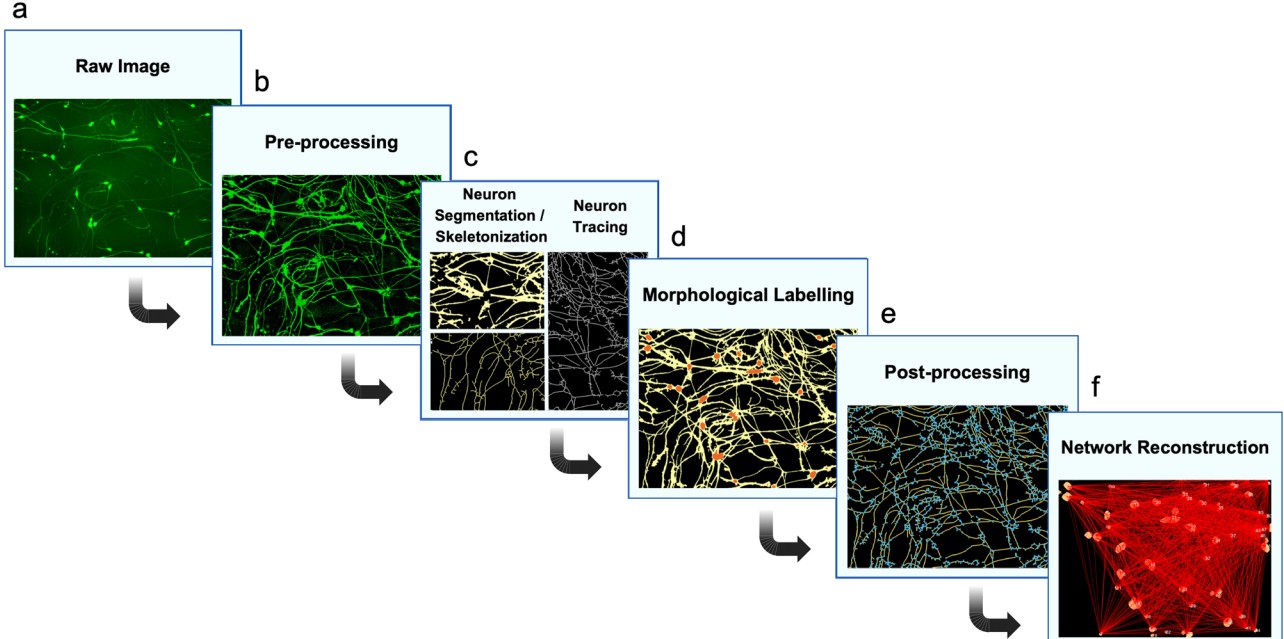

internodal connectivity[24]. This has since been echoed by other invertebrate and vertebrate neuron culture studies[24,26–28].

No reviews thus far have documented available tools to support research in structural in vitro neuronal network mapping. Instead, previous reviews have examined software that performs one facet of network reconstruction—neuron segmentation—particularly with a focus on 3D microscopy images[29–33]. Those with a broader scope have covered 2D as well as 3D segmentation tools, including excellent papers by refs. 34–37.

The current review offers a practical guide to assist cellular and network neuroscientists in selecting the most appropriate tools to quantify neuron cultures through network science. We first provide a conceptual overview of the steps involved in reconstructing networks from microscopy images of 2D neuronal cultures, highlighting key challenges and decision junctures. We next review algorithms and ready-to-implement freeware for automatic network reconstruction, alongside non-technical accounts of their capabilities and functionality.

## The network reconstruction framework

A series of transformations are used to convert raw microscopy images into network graphs (Fig. 2). In this pipeline, the first step is image acquisition (Fig. 2a). Of all microscopy types, light microscopy is perhaps most relied upon for the visualization of neuronal culture due to its accessibility,

versatility, and relatively simple sample preparation requirements. The second step is digital preprocessing of raw images (Fig. 2b), which involves enhancing signal-to-noise ratio and removing extraneous debris. Third is segmentation or tracing (Fig. 2c). Segmentation demarcates structures of interest as a binary mask, and can be subsequently transformed into a skeleton. Tracing extracts a compact representation of neurites by characterizing only their centerlines. Fourth, morphological features such as neurites and somata are labelled according to intensity-, shape-, or texture-based criteria (Fig. 2d). Fifth is postprocessing of the segmentation in preparation for network reconstruction, which includes reparation of discontinuities, branch pruning, or extraction of finer morphological detail such as neurite branch points (Fig. 2e). Lastly, network reconstruction transforms the labelled segmentation into a graph comprised of nodes and edges (Fig. 2f). These graphical elements may represent somata and neurites, or any other neuromorphological relationships. Each of these steps is further explored in the following sections, and italicized items are defined in a glossary in Supplementary Material 1.

## Microscopic image acquisition

Two-dimensional neuron culture is compatible with a variety of high-volume microscopy applications, such as live-cell, high-throughput, and high-content imaging. In this context, fluorescence microscopy is often employed to reveal cellular and subcellular features with high signal-to-noise ratio and target specificity. A common technique for visualization is immunostaining, in which protein epitopes specific to certain neuron structures are tagged with fluorophore-conjugated antibodies[38]. Popular fluorescence microscopy systems include widefield, confocal, light sheet, two-photon, and super-resolution microscopy; all of which have been technically detailed in excellent reviews[39,40]. One significant consideration of immunostaining, however, is that the sample must be preserved with a fixative before stain application, which prevents analysis at future timepoints in the cellular lifespan. An alternative is the use of probes compatible with ongoing biological processes for longitudinal live-cell recording, such as fluorescent reporter proteins[41]. In all these applications, fluorescence imaging carries the risk of certain ambiguities that should ideally be mitigated later in computational processing pipelines. For example, a low signal-to-noise ratio can arise from background autofluorescence, out of focus light, or ineffective staining techniques[42]. Photobleaching, where the fluorophore loses its ability to fluoresce over time due to prolonged or repeated exposure to light, can similarly compromise signal[43].

Other types of light microscopy rely on intrinsic properties of the sample rather than exogenous fluorophores for visualization, and are thus highly suited to live-cell imaging applications. Brightfield microscopy is one example in which light is transmitted through a specimen and the generated optical properties are used to create images. This microscopy technique is time- and cost-effective, however lacks the ability to provide specific labelling of molecules or structures within a sample. In addition, it can be sensitive to the uneven illumination and scattering that commonly produce image artifacts and reduce contrast[44].

## Pre-processing

The goal of pre-processing is to maximize the likelihood that target structures in microscopy images are recognized by subsequent detection algorithms. Applicable techniques aim either to systematically correct distortion originating from microscopic acquisition, such as in the case of deconvolution[45], or to improve image clarity through discrete image transformations. For example, some approaches target uneven background illumination by equalizing or normalizing the range of pixel intensity values[46,47] or introducing spatial smoothing filters such as a *Gaussian* or *median blur*[3,48,49]. In addition, pre-processing methods exploit the fact that debris usually has a size and shape dissimilar to neurons by including *morphological opening* for their removal[50]. To facilitate the identification of structure boundaries in later stages, edge contrast can also be improved with *Laplacian high pass filters*[51]. Software tools often automatically apply a battery of these techniques, for example NeuriteTracer[52] performs contrast enhancement, *rolling ball background subtraction*, despeckling and *Gaussian blurring* to improve the signal of neuron structures (Supplementary Fig. 1).

## Neuron segmentation and tracing

Structures of interest must be separated from surrounding or irrelevant regions in order for reconstruction to take place. In this regard, two primary methods are used to discern neurons in microscopy images: segmentation or tracing. Segmentation aims to create a model of neurons that is representative of their morphological structure. Global thresholding is one employed segmentation technique that relies on a cutoff pixel intensity value to partition the entire image into foreground or background[53]. Early neuron reconstruction workflows[54,55] and some contemporary tools such as NeuriteTracer[52] require the user to manually select this numerical threshold, while other tools implement automatic thresholding algorithms. For example, the *Otsu algorithm* classifies pixels into foreground or background based on an optimal value that minimizes within-class and maximizes between-class variance[56]. Despite its merit, this approach can underperform in cases where foreground pixel intensities are better characterized by multiple classes rather than one. For this reason, tools that implement automatic *Otsu thresholding* often include options for user input through manual parameter adjustment[57] or selectable thresholding settings[58]. Other global algorithms include Huang's thresholding, based on fuzzy set theory[59], or maximum entropy[60,61] and Li's[62] thresholding, based on the entropy principle, although these are not featured prominently in tools covered by this review. In practice, global thresholding techniques may be limited in the context of significant variations in image intensity stemming from noise or uneven illumination. In such cases, adaptive thresholding, where dynamic cut-off values are calculated according to local pixel neighborhoods rather than a global threshold, can be more appropriate[63–65]. Tool pipelines such as ExplantAnalyzer incorporate user-driven methods to optimize the neighborhood window size, ensuring it is as small as possible while still remaining larger than the greatest neurite width[64]. Adaptive thresholding procedures, however, assume that the window size contains a sufficient number of foreground and background pixels to calculate an appropriate average intensity threshold. This may be infeasible in certain image datasets that contain expansive background regions unpopulated by cells, or in other cases, may require excessive tuning on the behalf of the user.

Segmentation approaches can also be based on discerning boundaries between foreground and background objects. Here, gradient analysis identifies edges of the neuron by rapid changes in intensity[66–70]. To avoid the fragmentation of edge pixels, the extraction and linkage of edge orientation fields can build continuous contours along the boundary of neurite filaments[66]. Certain algorithms such as those employed by NeurphologyJ[71] (Supplementary Fig. 2c, d) and GAIN[57] (Supplementary Fig. 2e, f) combine user-parametrized intensity thresholding and edge detection to maximize the likelihood that thick and thin neurites are detected respectively.

After segmentation is performed, skeletonization may be implemented to compress the mask into a single pixel-wide structure, as employed by NeuriteTracer[52] (Supplementary Fig. 2a, b). Common algorithms for this purpose include medial axis transforms[50,72], which generate a skeleton at centerlines equidistant from object boundaries, and homotopic thinning[46], which generates a skeleton with preserved topological features. Alternatively, a one pixel-thick representation may be obtained through a process called tracing. This involves the iterative reconstruction of neurite centerlines direct from microscopy images according to local (and occasionally global) information. To achieve this, feature similarity between pixels guides directional kernels along midlines[45,47,73] or regions[74]. Tracing may also be framed as a graph problem, with pixels or nuclei as nodes. Here, edge weight confers the minimum cost path, which is used to produce a final tree structure[75–78]. Other approaches follow a probabilistic framework to strengthen tracing performance in ambiguous cases. For example, proposed *Bayesian frameworks* build evidence for a set of trace predictions by using a combination of current measurements and prior knowledge of geometric or intensity-based features[79,80]. Both skeleton and tracing representations are

ideal for quantifying geometrical features like neurite direction, length, and branching[71,81], and may be further refined in post-processing by techniques such as pruning. However, these complementary models do not explicitly consider morphological information such as shape or thickness, rendering them less suited to studies of neuroanatomy than segmentations.

While most reconstruction procedures such as adaptive and global thresholding rely solely on pixel intensity-based criteria, deep learning architectures account for other diverse contextual pixel features such as texture and shape to establish high-performing predictive frameworks[82–86]. This greatly enhances their ability to overcome poor contrast, fuzzy structure boundaries, and morphological heterogeneity[87]. For example, *convolutional neural network* (CNN) architectures such as residual networks[82] build progressively more complex feature maps to form a hierarchical representation of the target image. They have shown effectiveness in segmenting not only fluorescent microscopy images, but also phase contrast images that lack cell fluorescent markers[82–85]. Encoder-decoder networks are employed to a lesser extent in neuron reconstruction but exhibit similar utility due to their ability to compress and subsequentially reconstruct low-dimensional image features[88]. Self-supervised deep learning networks utilizing these architectures may be customized to distinct protocols using relatively small amounts of empirical training data after pretraining on open general databases[88]. Alternative supervised and semi-supervised approaches allow manual classifier training, and platforms such as NeuriTES[89] and a toolbox by ref. [90] make this process user-friendly by incorporating training phases at relevant pipeline steps. Despite their merit, the computational resources, amount of pretraining data, and level of user expertise required to develop and operate these architectures compared to traditional segmentation tools have likely contributed to their relative scarcity in the literature. Their notable adaptability to context-specific image ambiguities, however, justifies their continual refinement by future research.

## Morphological labelling

Once neuron structures are segmented from background, more nuanced cellular features may be extracted for subsequent assignment to nodes and edges. Somata and neurites are two such structures important to examinations of cell number, type and connectivity. Numerous pipelines[3,50] including NeuriteTracer[52] (Supplementary Fig. 3a–d) and GAIN[57] (Supplementary Fig. 3i, j) require multichannel images that include a nuclei stain in order to define somata masks. Alternative computational approaches such as that of NeurphologyJ[71] (Supplementary Fig. 3e–h) discern somata and neurite labels automatically without relying on immunostaining gathered at image acquisition. In this tool, morphological filters such as opening provide a way to isolate cell bodies by removing small structures including filamentous neurites. The brightness of somata relative to neurites has also been utilized to label these structures in tools such as NeuriteIQ[91] and WIS-Neuromath[92]. Once cell bodies have been defined, their subtraction from full neuron masks reveals full neurite masks[50,71,93].

## Post-processing

Post-processing further refines neuron reconstruction to aid interpretation. After tracing or skeletonization, inaccurate or extraneous branches can be removed with selective pruning[48,50] and any gaps caused by inhomogeneous staining repaired with break linking algorithms[45,49,74,94]. Furthermore, branching complexity is often explored through the extraction of neurite attachment points (somata-neurite intersection points) and end points (neurite terminal points). As adopted by NeurphologyJ[71], morphological dilation and erosion may be used to detect these points respectively (Supplementary Fig. 4a, b). Furthermore, some pipelines such as GAIN[57] (Supplementary Fig. 4c, d) individuate neurites at junctions by joining ingoing and outgoing branches together based on continuity in orientation[57,72] or other geometrically logical rules[72].

## Network reconstruction

Mapping a network object from neuronal connectivity enables quantification through graph theoretical analysis. To date, this goal has primarily driven microscale connectomic research in the ex vivo domain, with full or partial reconstructions of animal nervous systems established with high resolution microscopy techniques. Electron microscopy, for instance, images nanometer-thick tissue slices that can subsequentially be consolidated into cubic sections capturing fine cell and gap junction data. This was used in pioneering studies on nematode species to define classes of neurons based on morphological and connectivity profiles[95,96], and highlight implications for complex functions such as mating[97] and feeding behaviors[98]. Comparable 3D reconstructions of the Drosophila melanogaster brain with light microscopy yielded valuable resources such as the Virtual Fly Brain[99], which resolved the interconnectivity of 41 local processing units. A complementary analysis of global network properties found small world attributes and a hierarchical structure consisting of functionally segregated modules and submodules[100]. In vertebrates, the Allen Mouse Brain Atlas[101] was established as a mesoscale weighted connectome based on axon volume between grey matter regions. Graphical analysis revealed a high number of hubs and a large clustering coefficient, in essence showing mixed properties of small world and scale free networks. Indeed, characterizing neurons within a living system – their arborization in 3D space, association with non-neuronal cells, and regional patterning – assists in contextualizing the mechanisms driving network organization in isolated in vitro environments.

Meaningful organisation is embedded at every scale of neuronal culture, and network representations accordingly capture biological data at different levels of dimensionality. At the smallest scale, axon and dendrite dynamics may be examined through graphs that model neurite *branch-* and *end-points* as nodes, and neurites as edges[25,64]. Corresponding graph reconstruction algorithms usually require a neurite tracing or skeleton as input. The skeleton is iteratively traversed to establish graphical elements based on local pixel neighbourhoods; *branch point* nodes by the presence of two adjoining pixels, and *end point* nodes by the presence of exactly one[102]. These methods have been used to study radial neurite outgrowth in high resolution microscopy datasets[64], and would also be highly applicable to other investigations of synaptogenesis, neuritogenesis, and axonal fasciculation in pathogenic systems[103]. At larger scales, nodes are typically assigned to cell landmarks established in morphological labelling steps. Existing graph-building routines represent nodes as individual cell bodies and edges as neurites[26,90], which would be ideal for analyses of sparse or dissociated cultures. Other routines represent somata clusters as nodes, which is highly applicable to mature cultures where cells tend to display collective organisation into mesoscopic structures[25,26]. These larger scale reconstructions stand to greatly enrich investigations of structure-function coupling, where they could provide an ideal complement to electrophysiological data in examining organised multimodal dynamics over time[21,22,104]. Lastly, some definitions of connectivity rely on spatial rather than anatomical relationships. *Euclidean*-based distance metrics distinguish cells that lie within close proximity, which has been utilised to produce graphs illustrative of local community structure and associated cell-cell interactions[105]. Only select methods have considered edge weight, by using measures of neurite length[23,64], and few have introduced edge direction. These represent compelling areas for future research.

Graph extraction is a quantitative mapping problem at its core and thus amenable to more generic algorithms than neuron segmentation. For example, the *Skel2Graph3D algorithm*[102] was originally developed outside of neuroscience, yet effectively constructs networks featuring neurite *branch/ end points* as nodes and neurites as edges (Supplementary Fig. 5). In graph building, anatomical connectivity is typically established in three steps: the neurite mask without nodes is morphologically dilated, the node structure mask is superimposed, and connected binodal paths are extracted as edges for the resulting graph output[86]. Graph pruning algorithms may remove extraneous paths from a network to simplify its structure. For example, retaining only the shortest path between two key nodes such as *attachment-* and *end-points* isolates routes that most likely inform efficient signal propagation[64]. Streamlining networks in this way facilitates various downstream graph-related tasks such as pathfinding and the extraction of graph metrics.

## Available tools for neuron reconstruction

Processing tools for 2D cell assay reconstruction require different levels of user intervention: ranging from the semi-automatic tracing methods of NeuronJ[106] and Simple Neurite Tracer[107], to fully automatic global reconstructions[34]. The efficiency of automatic platforms that lends them so distinctly to high-volume applications unfortunately also reduces their versatility, such that each tool performs optimally with certain subsets of input microscopy data. For this reason, the current review examines the specifications of automatic reconstruction tools, in particular those that are readily available as open access code, plugins or GUI implementations. Additionally, software reconstructing both morphological and graph models are considered based on the aforementioned importance of segmentation in network assembly.

Table 1 summarizes the input requirements and capabilities of each tool, while Table 2 summarizes their metric readouts. The following sections 3.1 and 3.2 highlight key algorithmic approaches employed by each program.

## Segmentation and tracing tools

Available segmentation tools are diverse and include commercial software such as HCA-Vision (CSIRO Biotech Imaging)[48] and Neurolucida (MBF Bioscience)[108], as well as a broad repertoire of freeware that will serve as the focus of this review (Fig. 3). As a whole, the field of neuron reconstruction has seen a shift away from simple intensity-based thresholding to segmentation routines that incorporate more group-level pixel features such as object size, shape, and gradient. These sophisticated techniques better serve downstream network reconstruction algorithms in the assignment of node and edge identities.

The tools documented here were selected based on their ability to produce two outputs: a 2D neuron/neurite reconstruction and a set of quantitative metrics. The corresponding resources of these tools are presented in Table 3.

- AutoNeuriteJ[109] is an ImageJ plugin capable of analyzing individual cell morphology in dissociated cultures. This tool was developed to quantify neurons in early differentiation, and as such provides indicators of maturity such as axodendritic neurite classification. After preprocessing to homogenize background and enhance neurite signal, both neuron and nuclei channels are binarized with global thresholding. A series of images are produced depicting segmentations and skeletons of each neuron, as well as a text file containing neurite morphological information. It should be noted that AutoNeuriteJ is not suitable for dense cultures with overlapping neurites, and thus may have limited utility in building networks from population-level data.

- GAIN[57] is a MATLAB-based algorithm with interactive GUI that has the capability to resolve morphology on a cell-by-cell basis. Neuron and nucleus image pairs are first segmented with edge detection via *Sobel filtering* and double *Otsu thresholding* to minimize the effect of intensity variation between cell bodies and neurites. The neurite mask is skeletonized, and neurite branches on either side of junction points are paired into continuous filaments based on congruent angular orientation and pixel adjacency. This process individuates neurites to map whole single cells, which is especially applicable to network reconstructions denoting individual neurites as edges. The authors specify that this capability may be compromised in cases where neurites sharply change direction near junction points.

- MorphoNeuroNet[110] is an ImageJ plugin that is optimized for dense neurite cultures grown for more than 10 days. It is one of the few tools that segments individual somata from clusters, achieved by an *adaptive region growing algorithm* that uses nuclei in nucleus-stained images. To generate the neurite mask, *rolling ball background subtraction* prepares images for the generation of high intensity, unsharp, and *Laplacian filter* masks. These three masks are combined to create the final segmentation and skeleton. It should be noted that this tool only individuates somata and not neurites.

- NeuriTES[89] is a deep learning platform that is novel in its combined ability to segment brightfield images of neurons and track their evolution over time. After user-driven pretraining, images undergo adaptive contrast enhancement and segmentation via a *convolutional neuronal network model*. This segmentation is used to further characterize cellular features, how they change over time, and to what degree they are associated with particular biological processes via *transfer entropy*. For example, the authors found that cultured neuronal populations overexpressing an amyotrophic lateral sclerosis-linked mutation displayed alterations in the neurite attributes thickness, flatness, length, and number[89]. NeuriTES centers on processing neurons across the temporal domain and thus has limited scope to analyze spatially complex cultures.

- Neurient[111] is a MATLAB-based algorithm that traces and quantifies the degree to which neurites exhibit spatial alignment in dense populations. The routine computes orientation information for each neurite, as well as seed points at neurite maxima along centerlines; both of which subsequently serve a local exploratory tracing algorithm that produces a full tree representation. The output describes angular features of neurites that quantify neuronal alignment, however it should be noted that this tool does not segment somata and thus would be restricted to graphical interpretations with nodes as *branch-* or *end-points*.

- Neurite Analyzer[112] is a Fiji plugin that was developed to quantify neuritogenesis throughout neuronal differentiation. With nuclei and neuron images as input, morphological filters such as *Frangi's filter*[113] and the *Grayscale Morphology filter*[114] are used to establish neurite and somata masks respectively. Segmented masks are fortified with a hole-filling function before being skeletonized. An option exists to optimize the reconstruction for high-density neuron populations by accounting for cell aggregation, however, the authors note that the number of neurites per cell may be overestimated due to the tool's inability to discern neurite origin points from terminal points.

- NeuriteIQ[91] is a pipeline with GUI designed to process high-density cultures. In the workflow, fluorescent signal is first enhanced through *top-hat* and *bottom-hat transformations*. To label somata, regions with high pixel intensity are correlated with nuclei structures in the respective image. The employed neurite tracing technique identifies center points and extracts their associated local directions within a given field, which are then connected to form a continuous curvilinear structure. Extremely thin neurites down to the width of one pixel are also detected by a procedure that employs *non-maximum suppression* to remove extraneous pixels followed by a *hysteresis linking technique*. It should be noted that in batch processing, full reconstructions are exchanged for labelling and measurement metrics outputted to an Excel spreadsheet.

- NeuriteQuant[93] is an ImageJ macro established to process more developed cultures with long and intersecting neurites. Rather than solely relying on intensity for thresholding, this platform selectively enhances neurite and somata structures with shape-based analysis facilitated by the *Grayscale Morphology filter*[114]. After the neurite ensemble is skeletonized, the reconstruction as well as outputs are presented in a web-based data browser. In addition to overall metrics, averaged metrics per cell and per field are calculated. However, the authors acknowledge that per field metrics might be affected by the state of the culture at imaging, for example, sparse cultures may not have enough neurites in certain regions to enable accurate averaging.

- NeuriteSegmentation[63] is an ImageJ macro created for the segmentation of neurite outgrowth from spinal cord slice cultures and dorsal root ganglion cultures. Processing employs local adaptive thresholding based on the *Per Object Ellipse fit* method[115], which is optimized to integrate object size and shape into the binarization process rather than just signal intensity. This assists in detecting faint neurite structures and discounting artifacts in brightfield images. Notably, the algorithm was

**Table 1 | Input and output specifications of neuron segmentation and network reconstruction tools**

| Software Tool | Required Inputs | | Software Capabilities | | | | Neurite Segmentation | | Somata Segmentation | | Network Graph Reconstruction |
|---|---|---|---|---|---|---|---|---|---|---|---|
| | Microscopy image type | Nuclei and neuron image pair | Signal: noise enhancement[a] | Repair of fragmentary neurite signal[b] | Batch processing mode | Processing of overlapping neurites | For total population | For individual cells | For total population | For individual cells | |
| AutoNeuriteJ | Fluorescence | ✓ | | | | | | ✓ | | ✓ | |
| GAIN | Fluorescence | ✓ | ✓ | ✓ | ✓ | ✓ | ✓ | ✓ | ✓ | ✓ | |
| MorphoNeuroNet | Fluorescence | ✓ | ✓ | | ✓ | ✓ | ✓ | | ✓ | ✓ | |
| NeurITES | Brightfield | | ✓ | ✓ | c | ✓ | ✓ | | ✓ | | |
| Neurient | Fluorescence | ✓ | ✓ | | | ✓ | ✓ | | | | |
| Neurite Analyser | Fluorescence | ✓ | | ✓ | ✓ | ✓ | ✓ | | ✓ | | |
| NeuriteIQ | Fluorescence | | ✓ | ✓ | ✓ | ✓ | | | ✓ | | |
| NeuriteQuant | Fluorescence | | | | ✓ | ✓ | ✓ | | ✓ | | |
| NeuriteSegmentation | Brightfield | | | | ✓ | | ✓ | | ✓ | | |
| NeuriteTracer | Fluorescence | ✓ | ✓ | | ✓ | ✓ | ✓ | | ✓ | | |
| Neuron Image Analyzer | Fluorescence or brightfield | ✓ | ✓ | | | ✓ | ✓ | ✓ | ✓ | ✓ | |
| NeuronAnalyzer2D | Fluorescence | ✓ | ✓ | ✓ | ✓ | | | ✓ | | ✓ | |
| Neuroncyto II | Fluorescence | ✓ | ✓ | | ✓ | ✓ | ✓ | ✓ | ✓ | ✓ | |
| NeuronMetrics | Fluorescence | | ✓ | ✓ | ✓ | | ✓ | | | | |
| NEMO | Fluorescence or brightfield | | | | ✓ | ✓ | ✓ | | ✓ | | |
| NeuronRead | Fluorescence or brightfield | | ✓ | | | ✓ | ✓ | | ✓ | | |
| NeuroTreeTracer | Fluorescence | | ✓ | | | ✓ | ✓ | ✓ | ✓ | ✓ | |
| NeurphologyJ | Fluorescence | | ✓ | | ✓ | ✓ | ✓ | | ✓ | | |
| SynD | Fluorescence | | | ✓ | ✓ | ✓ | ✓ | | ✓ | | |
| WIS-Neuromath | Fluorescence | | | ✓ | ✓ | ✓ | | ✓ | ✓ | | |
| Cytonet | Fluorescence | | | | ✓ | | | | ✓ | | ✓ |
| DeepTEGINN | Fluorescence or brightfield | | ✓ | ✓ | | | ✓ | | ✓ | | ✓ |
| ExplantAnalyzer | Fluorescence | ✓ | ✓ | ✓ | ✓ | ✓ | ✓ | | ✓ | | ✓ |

[a]Signal:noise enhancement identifies tools that employ specialized background subtraction techniques beyond blurring or size filtering.
[b]Repair of fragmentary neurite signal identifies tools that employ operations or algorithms to bridge neurite gaps that have resulted from uneven staining, trans-planar localization of cells, etc.
[c]NeurITES performs batch processing on time-series images.

**Table 2 | Quantitative metric outputs of neuron segmentation and network reconstruction tools**

| Software Tool | Local Metrics | | | | | | Global Metrics | | | | | | | Other |
|---|---|---|---|---|---|---|---|---|---|---|---|---|---|---|
| | Neurite length per cell or cluster | Number of neurites per cell or cluster | Area of somata per cell or cluster | Hierarchal classification of neurite branches[a] | Spatial/morphological analysis of neurite branches | Sholl analysis[b] | Number of somata | Total length/area of neurites | Total somata area | Number of branch/end/attachment points | Average pixel intensity of neurons | Neuron polarity[c] | Graph metrics | |
| AutoNeuriteJ | ✓ | ✓ | ✓ | ✓ | | | ✓ | | | | | ✓ | | |
| GAIN | ✓ | ✓ | ✓ | ✓ | | | ✓ | | | | | | | |
| MorphoNeuroNet | | | | | | | ✓ | ✓ | ✓ | ✓ | | | | Somata shape |
| NeurITES | ✓ | ✓ | ✓ | ✓ | ✓ | | | ✓ | ✓ | | | | | |
| Neurient | | | | | | | | | | | | | | Angular alignment of neurites |
| Neurite Analyser | ✓ | ✓ | ✓ | ✓ | ✓ | | ✓ | ✓ | | ✓ | | | | Number of neurites |
| NeuriteIQ | ✓ | | | | | | ✓ | ✓ | ✓ | ✓ | ✓ | | | |
| NeuriteQuant | | | | | ✓ | | ✓ | ✓ | ✓ | ✓ | | | | |
| NeuriteSegmentation | | | | | | | | ✓ | | | | | | Distance of axon from somata |
| NeuriteTracer | | | | | | | ✓ | ✓ | | | | | | |
| Neuron Image Analyser | ✓ | | | | | | | | | | | | | Neurite orientation |
| NeuronAnalyzer2D | ✓ | ✓ | ✓ | ✓ | | | | | | ✓ | | | | Average neurite width, number of protrusions in growth cone |
| Neuroncyto II | ✓ | ✓ | ✓ | ✓ | | | ✓ | ✓ | | | | | | |
| NeuronMetrics | ✓ | ✓ | | | ✓ | | | ✓ | | ✓ | | ✓ | | Neuron territory |
| NEMO | ✓ | ✓ | ✓ | | ✓ | ✓ | ✓ | ✓ | | | ✓ | | | Fractal analysis, angular measurement |
| NeuronRead | | | | | | | ✓ | ✓ | ✓ | | | | | Somata shape and perimeter |
| NeuroTreeTracer | | | | | | | | | | | ✓ | | | |
| NeurphologyJ | | | | | | | ✓ | ✓ | ✓ | ✓ | | | | |
| SynD | | | | | | ✓ | ✓ | ✓ | ✓ | | ✓ | | | Synapse analysis, soma axis |
| WIS-Neuromath | ✓ | ✓ | ✓ | ✓ | ✓ | | ✓ | ✓ | | | ✓ | | | |
| Cytonet | | | | | | | | | | | | | ✓ | |
| DeepTEGINN | | | | | | | | | | | | | ✓ | |
| ExplantAnalyzer | | | | | | ✓ | | ✓ | | ✓ | | ✓ | ✓ | Convex hull enclosing neurites, neurite orientation |

[a]Hierarchical classification of neurite branches includes primary/secondary/tertiary or short/long or axon/dendrite classifications.
[b]Sholl analysis quantifies the complexity and arborization of neurites extending from parent somata.
[c]Neuron polarity metrics characterize axodendritic properties of neurites.

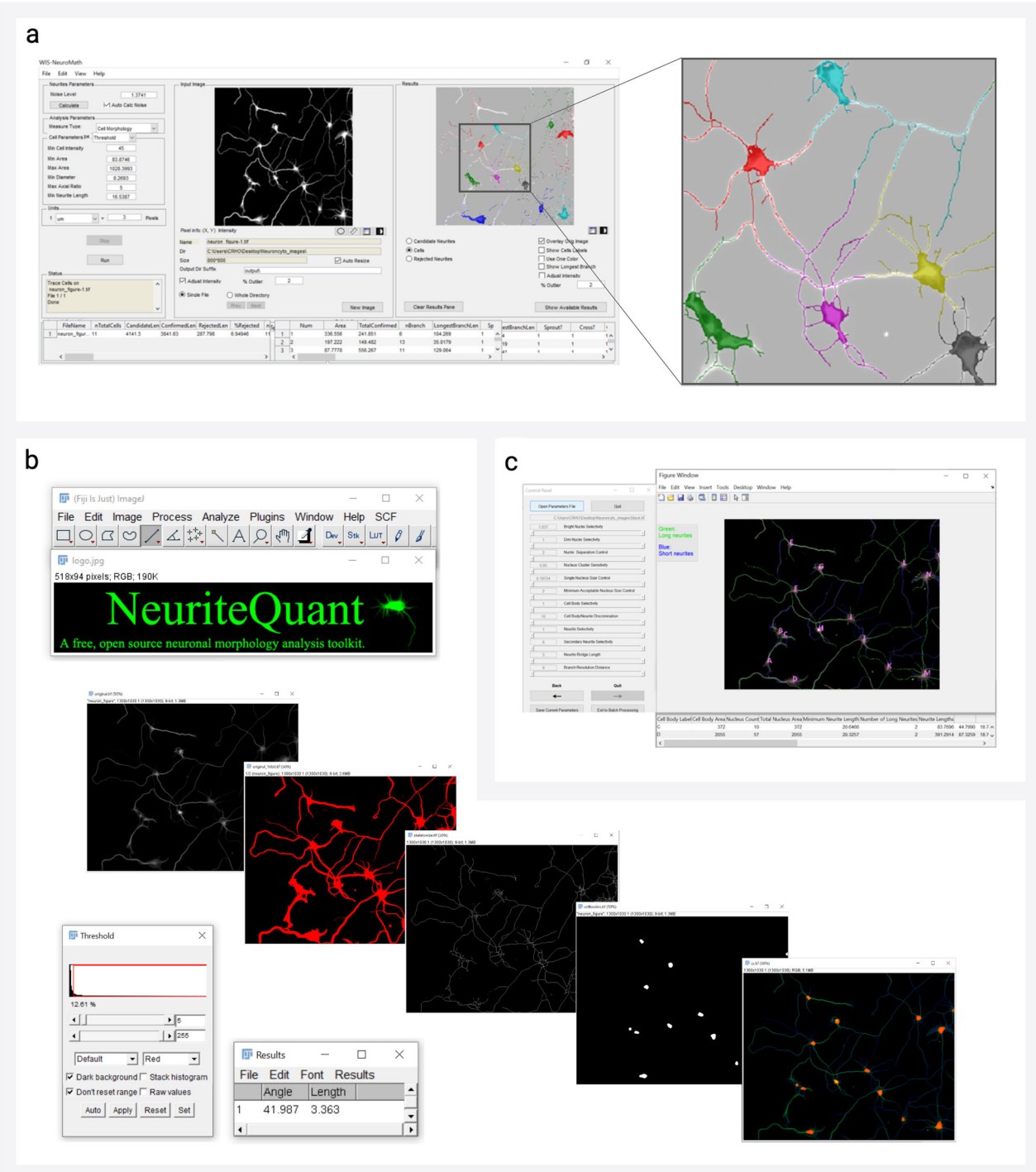

**Fig. 3 | Examples of neuron segmentation tool interfaces. a** WIS-Neuromath[92] graphical user interface with output depicting segmentation of individuated neurons, implemented in MATLAB[177]. **b** NeuriteQuant[93] interface with stages of neuron reconstruction from raw image to somata/neurite segmentation (CC BY 2.0), implemented in Fiji[178] (GNU General Public Licence). **c** GAIN[57] graphical user interface with output depicting segmentation of individuated neurons, implemented in MATLAB[177]. Neuron microscopy image utilized as input for tools sourced from Cell Image Library (CC BY 3.0)[179].

developed for segmentation from explant cultures and thus does not produce metrics that may be of interest in neuron culture analysis such as number of cell bodies.
- NeuriteTracer[52] is an ImageJ plugin designed to process fluorescence images of dissociated cultured neurons. After preprocessing to correct uneven illumination and enhance contrast, user-driven global thresholding of both nuclei and neurons is completed. Images are then despeckled with the Particle Remover plugin before a final neurite

skeleton is produced. One limitation noted by the authors of this platform is that it is unsuitable for dense neuron monocultures, and thus it may not be suitable for mesoscale network reconstruction pipelines.
- Neuron Image Analyzer[116] is a MATLAB-based tool that generates a vector representation of neurite structure as well as somata. Without nuclei stained images, soma are detected with a combination of a *Laplacian filtering* and *Level Set methods*. Neurite reconstruction relies

**Table 3 | Neuron segmentation software resources**

| Software | Authors | Interface | Installation Link |
|---|---|---|---|
| AutoNeuriteJ | Ref. 109 | ImageJ/Fiji | https://github.com/Grenoble-Institute-Neurosciences/AutoNeuriteJ |
| GAIN | Ref. 57 | Matlab-based GUI | https://github.com/qutublab/GAIN |
| MorphoNeuroNet | Ref. 110 | ImageJ/Fiji | http://www.limid.ugent.be/downloads.html[a] |
| NeuriTES | Ref. 89 | Matlab | https://github.com/Arianna1974/NeuriTES |
| Neurient | Ref.[111] | Matlab | https://github.com/jenmitch/neurient |
| Neurite Analyzer | Ref. 112 | ImageJ/Fiji | https://github.com/AlexisHaas/Neurite_Analyzer |
| NeuriteIQ | Ref. 91 | GUI | http://www.cbi-tmhs.org/NeuriteIQ/index.html[a] |
| NeuriteQuant | Ref. 93 | ImageJ/Fiji | http://ewit.ccb.tu-dortmund.de/groups/CB/bastiaens/dehmelt/NeuriteQuant/ |
| NeuriteSegmentation | Ref. 63 | ImageJ/Fiji | https://www.surgsci.uu.se/Forskning/forskningsomraden/Ortopedi/orto-lab-ikv/ |
| NeuriteTracer | Ref. 52 | ImageJ/Fiji | https://fournierlab.mcgill.ca/styled-6/NeuriteTracer.html |
| Neuron Image Analyzer | Ref. 116 | Matlab | https://github.com/kilho/NIA |
| NeuronAnalyzer2D | Ref. 67 | ImageJ/Fiji | https://mitobo.informatik.uni-halle.de/index.php/Applications/NeuronAnalyzer2D |
| NeuronCyto II | Ref. 117 | Matlab-based GUI | https://sites.google.com/site/neuroncyto/[a] |
| NeuronMetrics | Ref. 94 | ImageJ/Fiji | https://biii.eu/neuronmetrics[a] |
| NEMO | Ref. 58 | Matlab-based GUI | https://github.com/CentroEPiaggio/NEMO |
| NeuronRead | Ref. 118 | ImageJ/Fiji | Included as supplementary material in paper [126] |
| NeuroTreeTracer | Ref. 119 | Matlab | https://github.com/cihanbilge/AutomatedTreeStructureExtraction |
| NeurphologyJ | Ref. 71 | ImageJ/Fiji | https://hwangeric5.wixsite.com/erichwanglab/neurphologyj |
| SynD | Ref. 122 | Matlab-based GUI | software.incf.org/software/synd[a] |
| WIS-Neuromath | Ref. 92 | Matlab-based GUI | https://biii.eu/wis-neuromath |

[a]Website may not be actively maintained.

on relational pixel information established through a probabilistic *Hidden Markov Model*. Although this method significantly reduces the likelihood of off-target detection, it may also carry the risk of premature trace termination in cases where gaps in neurite staining exceed a $10 \times 10$ μm window. This tool also distinguishes axons from dendrites in highly arborized neurons with another probabilistic graph model that assesses information from entire neuronal tree in conjunction with local structure.

- NeuronAnalyzer2D[67] is an ImageJ plugin that reconstructs dissociated neurons and quantifies the distribution of subcellular fluorescently-labelled proteins. This tool is specialized to extract filopodia-like protrusions of the neurite growth cone, and as such, features multistep active contour models to capture fine morphology. Neuron structure is first binarized with the *Niblack thresholding method*, and a coarse contour is applied to detect approximate neurite regions. A second active contour model produces a more refined region edge through distance-based energy minimization. Finally, somata, neurite, and growth cone regions are definitively segmented by an algorithm that calculates the width profile along the structure. This tool focuses on elucidating microscale neuron morphology and thus may have a limited ability to process dense cultures at large fields of view.

- NeuronCyto II[117] is a MATLAB-based tool with GUI that features a novel technique to individuate neurites that are touching or intersecting. After stained neuron and nuclei images are provided as input, fluorescence signal is enhanced and noise is removed by preprocessing. Thresholding yields a binary segmentation, which is then overlaid with a trace to discern single neurites. The tracing process conceptualizes pixels as a directed graph and implements label propagation according to local and global contextual information[72]. The authors of NeuronCyto II highlight that occasional errors may occur in neurite individuation, which could be improved by incorporating metrics of width and brightness in future research. As this tool differentiates whole cells from clusters, it would be ideally suited to network reconstructions that distinguish edges as neurites and nodes as parent somata.

- NeuronMetrics[94] is an ImageJ plugin designed to process images of single neurons with complex neurite arbors. Its segmentation technique combines an *intensity threshold* mask to detect high-intensity neurites and a *Laplacian filter* mask to detect faint neurites, while the soma is selected manually. After neurite skeletonization, local exploratory methods are used for refinement, including a gap-filling algorithm that bridges broken neurite segments according to distance-based criteria. It should be noted that this tool segments only neurites and not somata.

- NEuronMOrphological analysis tool (NEMO)[58] is a MATLAB-based software with GUI optimized for batch processing and analysis of timelapse neuron microscopy images. Preprocessing may be performed manually or automatically and includes options for background homogenization and enhancing contrast. Segmentation is achieved through either grey level or *Otsu thresholding*, as well as edge detection. After skeletonization of neurites, if visualization of somata and neurites is preferred, the user must manually select each cell body. This is a limitation that may prevent application of this tool to larger datasets. However, NEMO is unique in its breadth of output metrics. Numerous readouts for each cell are collected in a data matrix, which forms the basis of feature extraction to reveal how phenotypes of cells differ over time and relative to one another. The tool also utilizes formal 3-way principal component analysis to determine statistically significant differences between cell, morphological metric, and timepoint data.

- NeuronRead[118] is an ImageJ macro that has the versatility to analyze both phase contrast and fluorescence images. Pre-processing procedures such as *median blurring* and *bottom-hat operations* are applied before a *watershed algorithm* separates clustered cell bodies. The authors note that this procedure may result in over-segmentation when cell body aggregates are present in the culture. For neurite segmentation, the image intensity histogram is adaptively equalized and *Difference of Gaussians filtering* is applied to enhance the edges of thin neurites, followed by skeletonization. This process is facilitated by comparison to a user-defined neurite width range.

- NeuroTreeTracer[119] is a MATLAB-based tool developed to individuate neurons in fluorescence images. A denoising algorithm[120] is first employed that reduces background fluorescence while preserving cell boundaries. Somata are then extracted using directional filters that detect local anisotropy. Neurite segmentation is performed with a machine learning approach based on *Support Vector Machines*[119,121], which notably, does require classifier training. For neuron individuation, neurite branches are conceptualized as graph trees, in which each node is connected to the root node (the somata) via a directed edge. Individuation at cell junctions is achieved by joining seed points with front-propagated traces based on neurite orientation. This tool assigns a unique label to each individuated neurite, which would streamline edge assignment in downstream network reconstruction routines.

- NeurphologyJ[71] is an ImageJ plugin targeted at reconstructing images produced by high-throughput screening. After standard preprocessing, various morphological operations are used to create a neuron mask. This segmentation is skeletonized, and a comprehensive point analysis is performed that includes the computation of a branching complexity metric summarizing neurite bifurcation. The authors noted that this tool may have a limited capability to skeletonize neurons at high magnification (≥ 40x) due to neurite diameter occupying more of the field of view. In network reconstruction applications, nodal assignments of either somata or *branch/end points* would be derivable from the output of this tool, potentially making it a good candidate for a multiscale graph pipelines.
- Synapse Detector (SynD)[122] is a MATLAB-based software with GUI for the detection of synapses and neurites. The image is preprocessed with an adaptive *Weiner filter* and globally thresholded, then somata are detected through *morphological opening*. Starting from the soma as seed points, neurite tracing is performed with steerable filters that calculate plausible directions of neurite ridges using a cost function. Two steerable filters with different sized filter kernels are applied to detect thick and thin neurites. To repair discontinuities in staining, the filter search radius is slightly extended from purported *end points*. Synapses are also identified based on unique local intensity maxima, which could assist in deriving functional connections for network reconstruction analysis. While this tool allows batch processing, the computational resources required for locally-driven neurite tracing can be more significant than global segmentation approaches.
- WIS-NeuroMath[92] is a MATLAB-based tool with GUI equipped to extract metrics from single cells in a population. Somata are first labelled based on simple intensity-based threshold segmentation. To identify candidate neurites, this tool employs an edge detection routine followed by a *stochastic completion*-like process. A final neurite skeleton is created by framing the trace as an undirected graph in which neurite pixels as treated as nodes. Edges are established between nodes along candidate neurite paths based on *Euclidean distance*. If neurite intersections occur, neurite lengths are assigned to respective cells based on an equidistant point. It should be noted that this method is not based morphological indicators such as neurite orientation, and thus can have some limitations in assigning correct neurites to somata in dense cultures.

## Extending segmentation tools: benchmarking methods

Given ongoing development since the 1980s, segmentation tools in cellular neuroscience have undergone extensive validation efforts that aim to reconcile biological subjects and computational reconstruction. Manual tracing remains the gold standard and is often used as a benchmark in quality assessment of automatic modelling algorithms. To facilitate this, large community platforms such as BigNeuron have been established that contain diverse light microscopy datasets and their corresponding gold standard annotations provided by human experts[123]. Powerful metrics have been proposed that enable comparison between these benchmark reconstructions and those produced by candidate automated segmentation methods. The DIADEM metric is one example that establishes similarity between reconstructions based on co-registration of bifurcations, terminal points, arborization patterns, and other criterea[124]. These resources accelerate the development of state-of-the-art tools by standardizing tests for algorithm evaluation as well as definitive target morphological outputs.

## Network reconstruction tools

A small number of existing toolboxes transform raw microscopy images into structural networks. Corresponding workflows go one step further than neuronal segmentation discussed in the previous section by establishing pixel-based criteria to resolve biological correlates of nodes and edges. These tools are documented below and recorded in Table 4.

- cytoNet[105] is a cloud-based platform with web interface that constructs networks from cell communities (Fig. 4a). It was developed to study spatial and functional relationships between neural progenitor cells with minimal neurite outgrowth[125]. The pipeline performs best with segmented images as input. However, the option to input raw microscopy images does also exist, as cytoNet is able to perform basic segmentation with *intensity thresholding* and *watershed operations*. Two types of spatial graphs may be generated, each with assignment of singular cells as nodes. Type I creates edges if the area of cells overlap after their mask boundaries are expanded by two pixels. Type II graphs create edges based on the proximity of cell centroids. To establish connectivity, it generates a threshold distance for each nuclei pair based on the average of the two object diameters, and multiplies it by a user-defined scaling factor. If the distance between the object centroid is lower than the threshold, an edge is established. Once networks have been constructed, local and global graph metrics are extracted to reveal neighbourhood characteristics. This tool was validated on human neural progenitor cells as they underwent differentiation, revealing an increase in clustering and the number of hub nodes by day 5[126].
- ExplantAnalyzer[64] is a MATLAB application designed to build weighted graphs from neurite trees of ex vivo tissue, especially organotypic explant cultures (Fig. 4b). A pair of images with a stained explant body nucleus and associated neurites are required as inputs. After standard pre-processing, neurite segmentation routines employ *adaptive thresholding*. The mask is *morphologically closed* to bridge gaps before being skeletonized. During graph reconstruction with the *Skel2Graph3D* function[102], nodes are demarcated as start points, *branch points*, and *end points*, and edges as connecting neurites. The adjacency matrix associated with this graph is weighted by the *Euclidean distance* between each node. The neurite graph is then reduced to a tree-like structure by only keeping the shortest path from each endpoint to start-point, determined by a backtracking algorithm. The authors note that this method could underestimate the morphological neurite length, as it allows edges to be part of more than one neurite tree if they are implicated in multiple shortest paths. In experimentation exploring the addition of neurotrophins on explant cultures, ExplantAnalyzer quantified a significant increase in neurite outgrowth and neurite *end points*, although a significant decrease in average shortest paths was not observed[64].
- A deep learning-based toolbox proposed by refs. 86,90. builds graphs from both brightfield and fluorescence microscopy images (Fig. 4c). By utilising an intuitive GUI, the user can construct a custom workflow from a library of methods that best suits their dataset. To prepare images for segmentation, standard pre-processing options may be selected from *OpenCV* and *SKimage* libraries. In addition, deep learning-based options can remove large and obfuscating artifacts, such as MEA electrodes, and estimate missing data in their place[127,128] Segmentation of structures can employ unsupervised methods, such as a modified *watershed algorithm*[129] or *W-Net*[130] model, or supervised methods that require labelling and training. The application of the deep learning algorithm yolov3 (You Only Look Once version 3.0[131]) categorises different brain cell types. This flexibility in nodal assignment could be highly applicable to multiscale graphs in which unique nodes are assigned to individual and clustered somata. After skeletonization with the *Zhang-Suen Thinning algorithm*[132], the core graph reconstruction process extracts *branch points* and *end points* as nodes, then assigns them cell type labels based on their proximity to yolov3-identified structures. Connectivity is determined by dilating the skeleton and establishing edges where the skeleton has binodal overlap. Edges belonging to non-neuronal cell types are then automatically removed based on the fact that they do not confer functional neuronal connections. It should be noted that the supervised learning-based components of the pipeline require the provision of training datasets. However, the toolbox offers a platform for user-driven generation of training data and explicitly defines the points at which this data must be integrated.

**Table 4 | Structural neuron network reconstruction resources**

| Software | Authors | Interface | Installation Link |
|---|---|---|---|
| cytoNET | Ref. 126 | Web browser | https://www.qutublab.org/how |
| Deep learning-based toolbox | Refs. 86,90 | Python-based GUI | https://github.com/gmorenomello/rfbi [a] |
| ExplantAnalyzer | Ref. 64 | Matlab | https://github.com/DominikSchmidbauer/ExplantAnlayzer |

[a]Website may not be actively maintained.

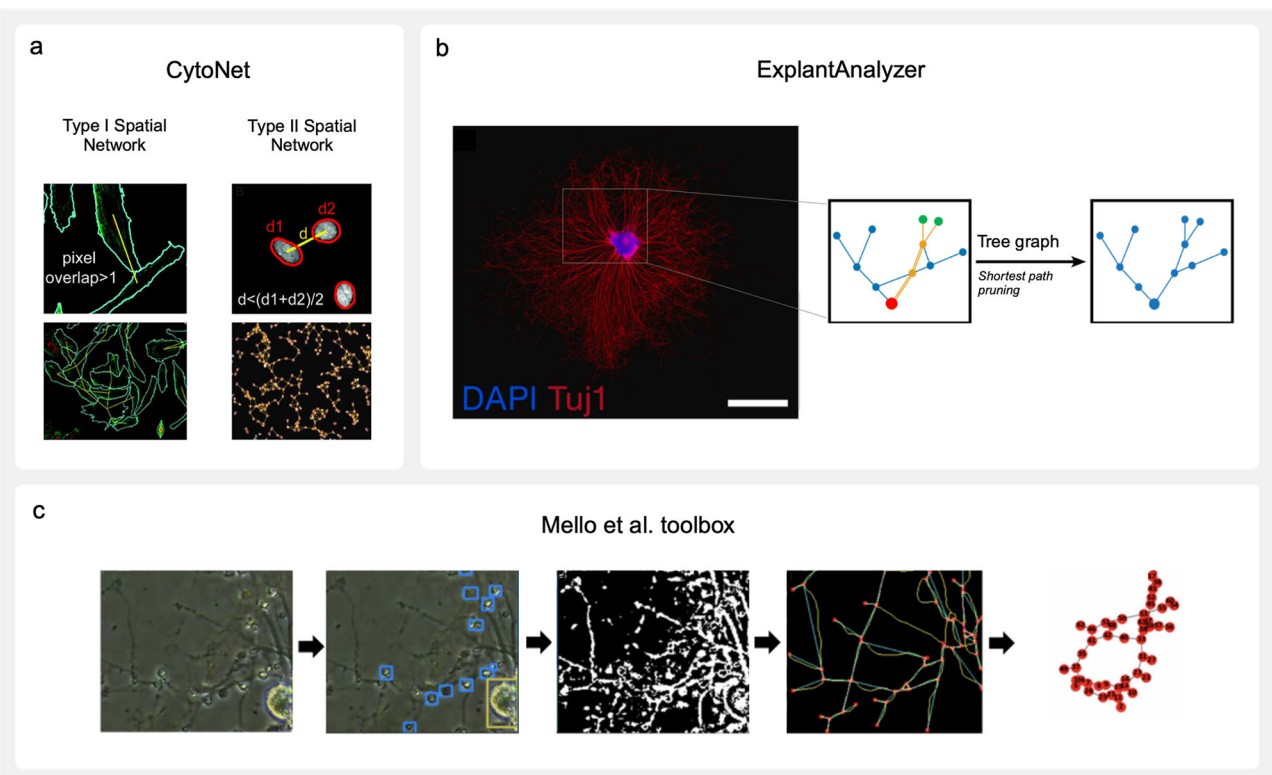

**Fig. 4 | Available network reconstruction software. a** Spatial network reconstructions by cytoNet[105]. Type I networks establish connections (yellow line) between cells (bordered in cyan) that touch when dilated. Type II networks establish connections (yellow line) between cells based on whether the distance between their nuclei (bordered in red) falls below a defined threshold. Image adapted from ref. 126 and modified (CC BY 4.0). **b** ExplantAnalyzer[64] network reconstruction of spiral ganglion explant neurites (stained with β-Tubulin III and DAPI, scale bar: 1 mm). A pruned graph structure is created by finding the shortest path from each neurite end point depicted in green to the explant body attachment point depicted in red. Edges not part of any shortest paths are removed from the final tree. Figure adapted from ref. 64 and modified (CC BY 4.0). **c** An example workflow of the network reconstruction tool developed by refs. 86,90. An input brightfield image of poor quality is morphologically labelled by yolov3; blue boxes are neuron somata, yellow box is a neuron somata cluster. After segmentation, a preliminary network is reconstructed with red nodes (branch and end points), blue edges, and yellow underlying skeleton, followed by final network reconstruction. Figure adapted from ref. 86 and modified with permission.

## Extending network reconstruction tools: benchmarking and analysis methods

Tool benchmarking procedures have not been systematically adopted in network reconstruction due to the nascency of the field, and thus represent a compelling opportunity for future research and platform development. One of the only exceptions is a study that developed a synthetic neuron image dataset with known network connectivity for validation purposes[25]. Here, synthetic images were created by first generating a connectivity matrix and then backfilling its edges into the image space with neurite segments isolated from real images. The network organization of synthetic images was then compared to that generated by the proposed algorithm using the F-measure – ranging from zero to one, where higher values indicate better predictive performance[25]. Alternative future methods could swap simulated neuron network analogues with physical counterparts whose structures are predetermined, intrinsically generating their own gold-standard graph constructs. For example, the technique of cell micropatterning utilizes specialty

plate biomaterials to guide neurite outgrowth, and could be used to produce anatomically defined cultures with known connections[133]. Alternatively, artificial microstructural models could be built from predefined templates, emulating the concept of phantoms in diffusion tensor imaging (DTI) where small-scale, synthetic brain models are made using fibers of polyester, rayon, or nylon to simulate white matter tracts[134–136]. After scanning to obtain DTI data, axonal fiber tractography is performed and the reconstructed phantom is compared to its known ground truth. The realization of physical neuron analogues would offer a self-contained system for validation where connectivity is designed at the outset and thus serves as an established ground-truth in later assessment.

In network analysis, measures such as the clustering coefficient and characteristic path length provide valuable insight into network topology. However, other mathematical tools offer a deeper exploration of the organizing principles underpinning observed complexity. Fractal analysis is one such example that explores the self-similarity of a system across scales by

detecting the prevalence of hierarchically repeating motifs[137–139]. Characterizations of in vitro neuron networks have revealed a wide multifractal spectrum that indicates high network heterogeneity, but shows increasing self-similarity over time at the level of mesoscale clustering[26]. Furthermore, geometric determinants of network organization can be examined through the construction of spatially embedded networks, which yield graph metrics that are meaningful in Euclidian space such as physical edge betweenness[140]. Network simulations known as generative models assist in elucidating the organizational features that emerge in biological systems as a result of spatiotemporal factors such as multifractality[137,141], alongside other neurodevelopmental constraints[142,143]. Additionally, examinations of communication dynamics elucidate the topological mechanisms that scaffold neuronal signaling across networks, providing a plausible bridge between structure and functional states[144]. Network control paradigms aim to determine which structural components drive system functionality through perturbative analysis[145,146], while other approaches model neuron spiking activity to reveal underlying topologies that could plausibly manifest emergent behavior[147]. Finally, resilience analysis may prove useful in quantifying the extent to which a network can withstand deterioration due to pathology associated with diseases such as schizophrenia or traumatic insult[148]. The majority of these network characterizations have centered on macroscale brain systems, however, future application to microscale neurocircuitry is also warranted. This will likely be accelerated by modern GPU-based parallel computing and emerging algorithmic architectures that illuminate properties of network data. For example, deep neural networks, which were previously unsuitable for graphical applications due to their non-Euclidian and inter-dependent nodal structure, have now been successfully adapted for these purposes[149]. Geometric deep learning architectures are able to process data features with inductive biases informed by geometric rules found in the physical world[150,151]. Neuroimaging studies have already harnessed this approach by encoding local space and spectral properties in geometric neural networks to uncover intrinsic features of functional connectivity[152]. On the other hand, topological deep learning architectures are ideally poised to encode higher-order relational properties. These models integrate principles from algebraic topology to learn complex global patterns in data[153–156]. Both geometric and topological deep learning have the potential to enrich future inference frameworks that aim to uncover inherent or empirically-relevant properties from neuron connectivity[157].

Despite the widespread adoption of network analysis in neuroscience, it is important to consider the limitations of graph modelling. Graphs intrinsically represent pairwise relationships such that one edge links two nodes only. While the study of topological motifs such as cliques offer some insight into higher-order interactions, neuronal relations are frequently collective in nature rather than confined to isolated patterns. Functional networks show synchronous firing of cells in triplicate, quadruplicate, and beyond[158], and it is easily discernible how this group-level connectivity would manifest structurally in underlying neurites that bifurcate or synapse with one another. Extensions to simple graphs are available to model simultaneous or heterogeneous connections between elements in a way that would not be captured by a traditional edge. For example, simplicial complexes are combinatorial structures that represent collections of geometric simplices and their relationships[159]. Wider mesoscale structures of simplicial representations are interpretable through the application of algebraic topology. These characterisations could offer insights pertinent to understanding information flow in neuronal ensembles, such as how cliques assemble to form higher-order cycles and cavities[160–163]. Quantifying persistent homology structure may reveal the robustness of such signatures in a neural system by uncovering their prevalence at different dimensional scales[164,165]. Furthermore, hypergraphs offer more generalised depictions of higher-order interactions[166]. These models have the capacity to depict three or more interactions between nodes through the inclusion of hyperedges. Applications in neuroscience have identified the expanded scope of representation that emerges from allowing a single modelled element to convey multiple biological connectivity scenarios[167]. This would have clear merit in recapitulating complex neuronal arborisation where several synaptic targets

can exist for a single source and vice versa. However, it is important to acknowledge that the increased generality of these models can translate to ambiguity in situations where precise relationships need to be defined in order to understand functional implications. For this reason, topological representations will most likely serve as valuable companions to network analysis in a specific constituent of experimental objectives.

## Future directions and concluding remarks

Systems-level analysis of neuronal architecture reveals subtle properties of cell viability and behavior that are not detectable though simple morphological analysis. Viewing these cells as networks provides a mathematical framework to quantify and analyze topological patterning through graph theory. Existing network reconstruction tools define either anatomical[64,90] or spatial[105] topology across multiple scales, highlighting the versatility of this framework in addressing diverse research questions. The widespread adoption of these tools in the cellular neuroscience community, however, will be contingent upon the strengthening of robust benchmarking methods that validate derived connectivity profiles. As many interpretations of connectivity are possible, reliable criteria and non-arbitrary cutoff points will be essential to establish at every phase of the reconstruction workflow. Foundational steps such as segmentation are well-supported in this regard by existing validation methods, however, novel benchmarking approaches based on simulated or physical ground truth models will be required in the future to standardize the quality of final network reconstruction.

In the body of available software, key algorithmic themes emerge at each stage of image processing. Initially, the primary focus of most pipelines is selective filtering to enhance signal-to-noise ratio and remove extraneous objects. To this end, spatial, morphological, and frequency filters adjust pixel values based on the surrounding features of their neighborhood. For subsequent segmentation and morphological labelling, several local and global approaches have been proposed. Joining the classic technique of intensity thresholding are novel procedures that incorporate shape-based criteria to optimize neuron detection, as well as local methods that trace centerline paths iteratively to account for changing features. In addition, deep learning methods offer powerful avenues for robust segmentation. Future research could prioritize interactive functionality in the development of supervised and semi-supervised methods, where manual annotation from the user is employed to shape and correct ground truth masks in real time. In this way, human expertise may be leveraged to resolve neuromorphological ambiguities alongside dataset-specific noise and idiosyncrasies. Investment in these methods in the wider sphere of generic cell segmentation has already enabled cell biologists without expertise in computer science to quantify their datasets in an intuitive, flexible, and robust manner, as illustrated by the tools Ilastik[168], Trainable Weka-Segmentation[169] and LABKIT[170]. These platforms, however, are not designed to provide specific neuromorphic readouts.

Segmentations can be refined to yield graphically meaningful features destined for network node and edge identities. However, a limited capacity to resolve intricate neuronal ensembles poses an issue for network reconstruction algorithms that base connectivity on discrete structures. This challenge likely underlies the sparsity of neuron network reconstruction in the literature. After all, many domain-agnostic algorithms have been proposed for the reconstruction of filamentous patterns, such as leaf venation, slime mold populations, and mud cracks[171,172]. If this is the case, the key to expanding neuron-specific resources lies not in developing more graphical model approaches per se, but rather in adding more graphically-relevant features to segmentation routines. Node and edge localization as well as edge weight and direction may be facilitated by more biologically relevant quantifications of diameter, size, orientation, pixel intensity, and so forth. Realizations of this already exist in macroscale brain network reconstruction. For example, numerous DTI schemes leverage proxies for cross-sectional width or number of axonal streamlines connecting brain regions to inform edge weight in network models[173]. Streamline filtering methods, such as SIFT[174,175] and COMMITT[176], have been developed to yield connectivity measures that are more consistent with underlying white matter

ultrastructure than simple streamline counts. Equivalent formulations at the microscale might extract the number of inter-somal neurites or the thickness of fascicular bundles from segmented reconstructions to inform the same target. Further, morphological features of neurites such as length and arborization may provide a basis for axonal or dendritic assignment, which could ideally serve the introduction of edge directionality. This would be facilitated by imaging modalities such as confocal and phase contrast microscopy, whose high resolution and scanning field cater to cell visualization at both ultrastructural and population levels. It is desirable for network reconstruction workflows to be optimized from these early stages to ensure a well-integrated approach to target research objectives.

Overall, the prospective of applying graph theoretical analysis to neuronal networks is one of great significance in exploring microscale neuromorphology and organization. Combining graph theoretical approaches and advanced segmentation techniques stands to greatly enrich our understanding of neuronal microcircuitry and pave the way for new discoveries in the field of cellular neuroscience.

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

## Acknowledgements
This research was supported by an Australian Government Research Training Program (RTP) Scholarship.

## Author contributions
Conceptualization, C.H., E.C., M.D.B., A.Z.; Investigation, C.H. and E.C.; Writing – Original Draft, C.H.; Writing – Review and Editing, C.H., M.D.B., E.C., A.Z.; Supervision, M.D.B., A.Z., E.C.; Visualization, C.H. and E.C.

## Competing interests
The authors declare no competing interests.
