## [Peer Review File · Communications Biology]

Reviewers' comments:

Reviewer #1 (Remarks to the Author):

In this manuscript, the authors give a brief review of the analysis and modelling of microscopy images of 2D neuronal cultures. They focus on the construction of network models, spanning from data pre-processing, neuron segmentation and skeletonization, neuron tracing, morphological labelling, post-processing, network model construction, and network analysis and modelling. They have covered not only various image-processing steps and procedures, but also algorithms and existing software for implementing them.

This review paper is very solid, logic, and well-written. The topics are of great interests and importance. I would recommend the publication with only minor revision.

My major concern is the network analysis and modelling part. In fact, it has been widely known that networks or graphs are suitable for pair-wise interactions and tend to miss the higher-order interactions. Recently, simplicial complexes, hypergraphs, and other topological representations have demonstrated a great potential in the analysis of intrinsic interactions within various data (including neural data). For instance,

Giusti, Chad, Robert Ghrist, and Danielle S. Bassett. "Two's company, three (or more) is a simplex: Algebraic-topological tools for understanding higher-order structure in neural data." *Journal of computational neuroscience* 41 (2016): 1-14.

Battiston F, Cencetti G, Iacopini I, Latora V, Lucas M, Patania A, Young JG, Petri G. Networks beyond pairwise interactions: Structure and dynamics. *Physics Reports*. 2020 Aug 25;874:1-92.

Otter, N., Porter, M.A., Tillmann, U., Grindrod, P. and Harrington, H.A., 2017. A roadmap for the computation of persistent homology. *EPJ Data Science*, 6, pp.1-38.

Sizemore AE, Phillips-Cremens JE, Ghrist R, Bassett DS. The importance of the whole: topological data analysis for the network neuroscientist. *Network Neuroscience*. 2019 Jul 1;3(3):656-73.

Further, network-based deep learning models such as geometric deep learning and topological deep learning will have great potential for revealing the intrinsic properties of neural network data. For instance,

Bronstein MM, Bruna J, LeCun Y, Szlam A, Vandergheynst P. Geometric deep learning: going beyond euclidean data. *IEEE Signal Processing Magazine*. 2017 Jul 11;34(4):18-42.

Bronstein MM, Bruna J, Cohen T, Veličković P. Geometric deep learning: Grids, groups, graphs, geodesics, and gauges. *arXiv preprint arXiv:2104.13478*. 2021 Apr 27.

Chi Seng Pun, Si Xian Lee, and Kelin Xia, "Persistent-homology-based machine learning: a survey and a comparative study." *Artificial Intelligence Review*, 55, 5169–5213, (2022)

Hajij M, Zamzmi G, Papamarkou T, Miolane N, Guzmán-Sáenz A, Ramamurthy KN, Birdal T, Dey TK, Mukherjee S, Samaga SN, Livesay N. Topological Deep Learning: Going Beyond Graph Data.

I would recommend the authors to add some more discussions regarding to these new developments.

Reviewer #2 (Remarks to the Author):

This work highlights the significance of 2D neuron cultures in neuroscience, pointing out limitations in traditional metrics. It proposes interpreting in vitro neuron ensembles through network graphs to reveal energy and information transfer patterns. The study explores the application of network science to microscale neural systems, emphasizing a gap in tools for structural mapping of in vitro neuronal networks. The review aims to provide a practical guide for neuroscientists, covering conceptual overviews, addressing challenges, and reviewing algorithms for automatic network reconstruction from microscopy images of 2D neuronal cultures.

Overall the effort is commendable, the topic will be of interest to the neuroscience, network science and many other research communities and I am supportive of this manuscript. In what following, I describe a few issues that require further attention and need to be addressed:

1. How can segmentation methods in microscopy images be optimized to address challenges related to noisy or unevenly illuminated images, and what role do adaptive thresholding techniques play in enhancing accuracy in such scenarios? If this cannot currently be addressed, then the authors should check existing work and highlight this as a potential future direction so that the research community is guided by this paper.

2. What are the advantages and limitations of different approaches, such as skeletonization and tracing, in quantifying geometrical features like neurite direction, length, and branching in neuron microscopy images? Additionally, how do deep learning architectures, particularly convolutional neural networks (CNNs) and encoder-decoder networks, contribute to improving the accuracy and efficiency of segmentation and tracing applications for diverse neuron morphologies? Which papers from the references use and do not use and for what reason?

3. How can benchmarking methods be strengthened to provide robust validation for the derived connectivity profiles of existing network reconstruction tools, considering the multitude of possible interpretations of connectivity in systems-level neuronal analysis? What criteria and non-arbitrary cutoff points are essential at each phase of the reconstruction workflow to ensure accurate and reliable results?

4. In addressing the challenge of sparsity in neuron network reconstruction, what graphically relevant features, such as diameter, size, orientation, and pixel intensity, can be incorporated into segmentation routines to enhance the resolution of intricate neuronal ensembles? How can the addition of these features facilitate more accurate node and edge localization, as well as edge weight and direction determination, ultimately advancing our understanding of microscale neuromorphology and organization through graph theoretical analysis?

5. A very important aspect is the network characterization and study of neuron cultures, so the paper has to refer to pioneering works such as "Hidden network generating rules from partially observed complex networks." *Communications Physics* 4, no. 1 (2021): 199; "Deciphering the generating rules and functionalities of complex networks." *Scientific reports* 11, no. 1 (2021): 22964; "The role of long-term power-law memory in controlling large-scale dynamical networks." *Scientific Reports* 13, no. 1 (2023): 19502; "Neuron particles capture network topology and behavior from single units." *bioRxiv* (2021): 2021-12. and discuss them at length so that the future researchers interested in this topic know from where to start.

Response to Reviewers

We are grateful to the editorial team and reviewers for taking the time to review our manuscript. Please find our detailed response to the reviewers' comments below.

Reviewer #1:

In this manuscript, the authors give a brief review of the analysis and modelling of microscopy images of 2D neuronal cultures. They focus on the construction of network models, spanning from data pre-processing, neuron segmentation and skeletonization, neuron tracing, morphological labelling, post-processing, network model construction, and network analysis and modelling. They have covered not only various image-processing steps and procedures, but also algorithms and existing software for implementing them.

This review paper is very solid, logic, and well-written. The topics are of great interests and importance. I would recommend the publication with only minor revision.

We thank the reviewer for their enthusiasm in our work and constructive feedback.

My major concern is the network analysis and modelling part. In fact, it has been widely known that networks or graphs are suitable for pair-wise interactions and tend to miss the higher-order interactions. Recently, simplicial complexes, hypergraphs, and other topological representations have demonstrated a great potential in the analysis of intrinsic interactions within various data (including neural data). For instance,

Giusti, Chad, Robert Ghrist, and Danielle S. Bassett. "Two's company, three (or more) is a simplex: Algebraic-topological tools for understanding higher-order structure in neural data." *Journal of computational neuroscience* 41 (2016): 1-14.

Battiston F, Cencetti G, Iacopini I, Latora V, Lucas M, Patania A, Young JG, Petri G. Networks beyond pairwise interactions: Structure and dynamics. *Physics Reports*. 2020 Aug 25;874:1-92.

Otter, N., Porter, M.A., Tillmann, U., Grindrod, P. and Harrington, H.A., 2017. A roadmap for the computation of persistent homology. *EPJ Data Science*, 6, pp.1-38.

Sizemore AE, Phillips-Cremins JE, Ghrist R, Bassett DS. The importance of the whole: topological data analysis for the network neuroscientist. *Network Neuroscience*. 2019 Jul 1;3(3):656-73.

Response: We agree with the reviewer that alternatives to pairwise graphical modelling are relevant to the scope of this review, particularly higher order representations as specified. We have now outlined the potential caveats of pairwise modelling and described other models that recapitulate collective structural interactions, with a particular focus on hypergraphs and simplicial

complexes. We have also summarised tools from algebraic topology such as persistent homology that characterise dominant patterns in simplicial representations.

Modifications to the manuscript

Extending Network Reconstruction Tools: Benchmarking and Analysis Methods (page 23-24):

“Despite the widespread adoption of network analysis in neuroscience, it is important to consider the limitations of graph modelling. Graphs intrinsically represent pairwise relationships such that one edge links two nodes only. While the study of topological motifs such as cliques offer some insight into higher-order interactions, neuronal relations are frequently collective in nature rather than confined to isolated patterns. Functional networks show synchronous firing of cells in triplicate, quadruplicate, and beyond¹, and it is easily discernible how this group-level connectivity would manifest structurally in underlying neurites that bifurcate or synapse with one another. Extensions to simple graphs are available to model simultaneous or heterogeneous connections between elements in a way that would not be captured by a traditional edge. For example, simplicial complexes are combinatorial structures that represent collections of geometric simplices and their relationships². Wider mesoscale structures of simplicial representations are interpretable through the application of algebraic topology. These characterisations could offer insights pertinent to understanding information flow in neuronal ensembles, such as how cliques assemble to form higher-order cycles and cavities³⁻⁶. Quantifying persistent homology structure may reveal the robustness of such signatures in a neural system by uncovering their prevalence at different dimensional scales^{7,8}. Furthermore, hypergraphs offer more generalised depictions of higher-order interactions⁹. These models have the capacity to depict three or more interactions between nodes through the inclusion of hyperedges. Applications in neuroscience have identified the expanded scope of representation that emerges from allowing a single modelled element to convey multiple biological connectivity scenarios¹⁰. This would have clear merit in recapitulating complex neuronal arborisation where several synaptic targets can exist for a single source and vice versa. However, it is important to acknowledge that the increased generality of these models can translate to ambiguity in situations where precise relationships need to be defined in order to understand functional implications. For this reason, topological representations will most likely serve as valuable companions to network analysis in a specific constituent of experimental objectives.”

Further, network-based deep learning models such as geometric deep learning and topological deep learning will have great potential for revealing the intrinsic properties of neural network data. For instance,

Bronstein MM, Bruna J, LeCun Y, Szlam A, Vandergheynst P. Geometric deep learning: going beyond euclidean data. *IEEE Signal Processing Magazine*. 2017 Jul 11;34(4):18-42.

Bronstein MM, Bruna J, Cohen T, Veličković P. Geometric deep learning: Grids, groups, graphs, geodesics, and gauges. *arXiv preprint arXiv:2104.13478*. 2021 Apr 27.

Chi Seng Pun, Si Xian Lee, and Kelin Xia, "Persistent-homology-based machine learning: a survey and a comparative study." *Artificial Intelligence Review*, 55, 5169–5213, (2022)

Hajj M, Zamzmi G, Papamarkou T, Miolane N, Guzmán-Sáenz A, Ramamurthy KN, Birdal T, Dey TK, Mukherjee S, Samaga SN, Livesay N. Topological Deep Learning: Going Beyond Graph Data.

I would recommend the authors to add some more discussions regarding to these new developments.

Response: We agree that recent developments in deep-learning methods enrich the analysis of network graphs and thus warrant discussion. We have now included an exploration of geometric and topological deep learning architectures, and highlighted their use in neuroimaging research on microscale and macroscale connectivity.

Modifications to the manuscript

Extending Network Reconstruction Tools: Analysis and Benchmarking Methods (page 23):

“The majority of these network characterizations have centered on macroscale brain systems, however, future application to microscale neurocircuitry is also warranted. This will likely be accelerated by modern GPU-based parallel computing and emerging algorithmic architectures that illuminate properties of network data. For example, deep neural networks, which were previously unsuitable for graphical applications due to their non-Euclidian and inter-dependent nodal structure, have now been successfully adapted for these purposes¹¹. Geometric deep learning architectures are able to process data features with inductive biases informed by geometric rules found in the physical world^{12,13}. Neuroimaging studies have already harnessed this approach by encoding local space and spectral properties in geometric neural networks to uncover intrinsic features of functional connectivity¹⁴. On the other hand, topological deep learning architectures are ideally poised to encode higher-order relational properties. These models integrate principles from algebraic topology to learn complex global patterns in data^{15–17}. Both geometric and topological deep learning have the potential to enrich future inference frameworks that aim to uncover inherent or empirically-relevant properties from neuron connectivity¹⁸.”

Reviewer #2:

This work highlights the significance of 2D neuron cultures in neuroscience, pointing out limitations in traditional metrics. It proposes interpreting in vitro neuron ensembles through network graphs to reveal energy and information transfer patterns. The study explores the application of network science to microscale neural systems, emphasizing a gap in tools for structural mapping of in vitro neuronal networks. The review aims to provide a practical guide for neuroscientists, covering conceptual overviews, addressing challenges, and reviewing algorithms for automatic network reconstruction from microscopy images of 2D neuronal cultures.

Overall the effort is commendable, the topic will be of interest to the neuroscience, network science and many other research communities and I am supportive of this manuscript.

We appreciate the reviewer's interest and thoughtful suggestions on our review.

In what following, I describe a few issues that require further attention and need to be addressed:

1. How can segmentation methods in microscopy images be optimized to address challenges related to noisy or unevenly illuminated images, and what role do adaptive thresholding techniques play in enhancing accuracy in such scenarios? If this cannot currently be addressed, then the authors should check existing work and highlight this as a potential future direction so that the research community is guided by this paper.

Response: We understand that the reviewer seeks clarification on segmentation approaches that are well positioned to overcome image ambiguities impacting current paradigms. Although adaptive thresholding techniques are more versatile than global thresholding techniques, certain limitations prevent their applicability to all datasets. The revised text better outlines these shortcomings and contextualises the performance of adaptive thresholding relative to other approaches such as machine learning segmentation frameworks.

Modifications to the manuscript

Neuron Segmentation and Tracing (page 7): "In practice, global thresholding techniques may be limited in the context of significant variations in image intensity stemming from noise or uneven illumination. In such cases, adaptive thresholding, where dynamic cut-off values are calculated according to local pixel neighborhoods rather than a global threshold, can be more appropriate¹⁹⁻²¹. Tool pipelines such as ExplantAnalyzer incorporate user-driven methods to optimize the neighborhood window size, ensuring it is as small as possible while still remaining larger than the greatest neurite width²⁰. Adaptive thresholding procedures, however, assume that the window size contains a sufficient number of foreground and background pixels to calculate an appropriate average intensity threshold. This may be infeasible in certain image datasets that contain expansive background regions unpopulated by cells, or in other cases, may require excessive tuning on the behalf of the user."

Neuron Segmentation and Tracing (page 8): "While most reconstruction procedures such as adaptive and global thresholding rely solely on pixel intensity-based criteria, deep learning architectures account for other diverse contextual pixel features such as texture and shape to establish high-performing predictive frameworks²²⁻²⁶. This greatly enhances their ability to overcome poor contrast, fuzzy structure boundaries, and morphological heterogeneity²⁷."

2. What are the advantages and limitations of different approaches, such as skeletonization and tracing, in quantifying geometrical features like neurite direction, length, and branching in neuron

microscopy images? Additionally, how do deep learning architectures, particularly convolutional neural networks (CNNs) and encoder-decoder networks, contribute to improving the accuracy and efficiency of segmentation and tracing applications for diverse neuron morphologies? Which papers from the references use and do not use and for what reason?

Response: We agree that the limitations of skeletonization and tracing could be better explored and also wish to highlight the similarity of their reconstructed products. We have now included empirical examples of both tracing and skeletonization that illustrate their equal suitability for geometric analysis in the revised manuscript, substantiating previously outlined advantages. In addition, we have more comprehensively overviewed deep learning architectures that may be applied to neuron-based images, and emphasised their high performance in resolving commonly encountered image ambiguities. However, most existing neuron-specific segmentation and network reconstruction tools still rely on global or adaptive thresholding, and thus we highlight the importance of future investment in deep learning techniques.

Modifications to the manuscript

Neuron Segmentation and Tracing (page 8): “Both skeleton and tracing representations are ideal for quantifying geometrical features like neurite direction, length, and branching^{28,29}, and may be further refined in post-processing by techniques such as pruning. However, these complementary models do not explicitly consider morphological information such as shape or thickness, rendering them less suited to studies of neuroanatomy than segmentations.”

Neuron Segmentation and Tracing (page 8): “While most reconstruction procedures such as adaptive and global thresholding rely solely on pixel intensity-based criteria, deep learning architectures account for other diverse contextual pixel features such as texture and shape to establish high-performing predictive frameworks^{22–26}. This greatly enhances their ability to overcome poor contrast, fuzzy structure boundaries, and morphological heterogeneity²⁷. For example, *convolutional neural network* (CNN) architectures such as residual networks²² build progressively more complex feature maps to form a hierarchical representation of the target image. They have shown effectiveness in segmenting not only fluorescent microscopy images, but also phase contrast images that lack cell fluorescent markers^{22–25}. Encoder-decoder networks are employed to a lesser extent in neuron reconstruction but exhibit similar utility due to their ability to compress and subsequently reconstruct low-dimensional image features³⁰. Self-supervised deep learning networks utilizing these architectures may be customized to distinct protocols using relatively small amounts of empirical training data after pretraining on open general databases³⁰. Alternative supervised and semi-supervised approaches allow manual classifier training, and platforms such as NeuriTES³¹ and a toolbox by Mello et al.³² make this process user-friendly by incorporating training phases at relevant pipeline steps. Despite their merit, the computational resources, amount of pretraining data, and level of user expertise required to develop and operate these architectures compared to traditional segmentation tools have likely contributed to their relative scarcity in the

literature. Their notable adaptability to context-specific image ambiguities, however, justifies their continual refinement by future research.”

3. How can benchmarking methods be strengthened to provide robust validation for the derived connectivity profiles of existing network reconstruction tools, considering the multitude of possible interpretations of connectivity in systems-level neuronal analysis? What criteria and non-arbitrary cut-off points are essential at each phase of the reconstruction workflow to ensure accurate and reliable results?

Response: In the revised manuscript, we summarise existing and prospective benchmarking procedures for segmentation and network reconstruction in newly dedicated sections. Validation methods for segmentation tools, including morphological criteria for benchmark comparison, are well-established in the literature. Benchmarking procedures for microscale network reconstruction tools are comparatively rare, and as such, we outline potential future validation methods centred on bioengineered constructs of graph connectivity.

Modifications to the manuscript

Extending Segmentation Tools: Benchmarking Methods (page 18-19): “Given ongoing development since the 1980s, segmentation tools in cellular neuroscience have undergone extensive validation efforts that aim to reconcile biological subjects and computational reconstruction. Manual tracing remains the gold standard and is often used as a benchmark in quality assessment of automatic modelling algorithms. To facilitate this, large community platforms such as BigNeuron have been established that contain diverse light microscopy datasets and their corresponding gold standard annotations provided by human experts³³. Powerful metrics have been proposed that enable comparison between these benchmark reconstructions and those produced by candidate automated segmentation methods. The DIADEM metric is one example that establishes similarity between reconstructions based on co-registration of bifurcations, terminal points, arborization patterns, and other criteria³⁴. These resources accelerate the development of state-of-the-art tools by standardizing tests for algorithm evaluation as well as definitive target morphological outputs.”

Extending Network Reconstruction Tools: Benchmarking and Analysis Methods (page 21-23): “Tool benchmarking procedures have not been systematically adopted in network reconstruction due to the nascency of the field, and thus represent a compelling opportunity for future research and platform development. One of the only exceptions is a study that developed a synthetic neuron image dataset with known network connectivity for validation purposes³⁵. Here, synthetic images were created by first generating a connectivity matrix and then backfilling its edges into the image space with neurite segments isolated from real images. The network organization of synthetic images was then compared to that generated by the proposed algorithm using the F-measure – ranging from zero to one, where higher values indicate better predictive performance²⁵. Alternative future methods could swap simulated neuron network analogues with physical counterparts whose structures are predetermined, intrinsically generating their own gold-standard graph constructs. For example, the

technique of cell micropatterning utilizes specialty plate biomaterials to guide neurite outgrowth, and could be used to produce anatomically defined cultures with known connections¹³³. Alternatively, artificial microstructural models could be built from predefined templates, emulating the concept of phantoms in diffusion weighted imaging (DWI) where small-scale, synthetic brain models are made using fibers of polyester, rayon, or nylon to simulate white matter tracts^{134–136}. After scanning to obtain DWI data, axonal fiber tractography is performed and the reconstructed phantom is compared to its known ground truth. The realization of physical neuron analogues would offer a self-contained system for validation where connectivity is designed at the outset and thus serves as an established ground-truth in later assessment.”

Future Directions and Concluding Remarks (page 24): “Foundational steps such as segmentation are well-supported in this regard by existing validation methods, however, novel benchmarking approaches based on simulated or physical ground truth models will be required in the future to standardize the quality of network reconstruction.”

4. In addressing the challenge of sparsity in neuron network reconstruction, what graphically relevant features, such as diameter, size, orientation, and pixel intensity, can be incorporated into segmentation routines to enhance the resolution of intricate neuronal ensembles? How can the addition of these features facilitate more accurate node and edge localization, as well as edge weight and direction determination, ultimately advancing our understanding of microscale neuromorphology and organization through graph theoretical analysis?

Response: We now present several prospective examples demonstrating how morphological features derived from segmentation could improve network reconstruction, specifically focusing on edge weight and direction.

Modifications to the manuscript

Future Directions and Concluding Remarks (page 24): “Node and edge localization as well as edge weight and direction may be facilitated by more biologically relevant quantifications of diameter, size, orientation, pixel intensity, and so forth. Realizations of this already exist in macroscale brain network reconstruction. For example, numerous DWI schemes leverage proxies for cross-sectional width or number of axonal streamlines connecting brain regions to inform edge weight in network models⁴⁰. Streamline filtering methods, such as SIFT^{41,42} and COMMIT⁴³, have been developed to yield connectivity measures that are more consistent with underlying white matter ultrastructure than simple streamline counts. Equivalent formulations at the microscale might extract the number of inter-somal neurites or the thickness of fascicular bundles from segmented reconstructions to inform the same target. Further, morphological features of neurites such as length and arborization may provide a basis for axonal or dendritic assignment, which could ideally serve the introduction of edge directionality. This would be facilitated by imaging modalities such as confocal and phase contrast microscopy, whose high resolution and scanning field cater to cell visualization at both ultrastructural and population levels. It is desirable for network reconstruction

workflows to be optimized from these early stages to ensure a well-integrated approach to target research objectives.”

5. A very important aspect is the network characterization and study of neuron cultures, so the paper has to refer to pioneering works such as "Hidden network generating rules from partially observed complex networks." *Communications Physics* 4, no. 1 (2021): 199; "Deciphering the generating rules and functionalities of complex networks." *Scientific reports* 11, no. 1 (2021): 22964; "The role of long-term power-law memory in controlling large-scale dynamical networks." *Scientific Reports* 13, no. 1 (2023): 19502; "Neuron particles capture network topology and behavior from single units." *bioRxiv* (2021): 2021-12. and discuss them at length so that the future researchers interested in this topic know from where to start.

Response: We acknowledge the importance of contextualising graph reconstruction in broader frameworks that capture higher-order network topology. We have now included a discussion of metrics and simulations developed to quantify multifractal characteristics, generative organisational rules, and other statistical properties of neuronal networks.

Modifications to the manuscript

Extending Network Reconstruction Tools: Benchmarking and Analysis Methods (page 23):

“In network analysis, measures such as the clustering coefficient and characteristic path length provide valuable insight into network topology. However, other mathematical tools offer a deeper exploration of the organizing principles underpinning observed complexity. Fractal analysis is one such example that explores the self-similarity of a system across scales by detecting the prevalence of hierarchically repeating motifs^{44–46}. Characterizations of *in vitro* neuron networks have revealed a wide multifractal spectrum that indicates high network heterogeneity, but shows increasing self-similarity over time at the level of mesoscale clustering⁴⁷. Furthermore, geometric informants of network organization can be examined through the construction of spatially embedded networks, which yield graph metrics that are meaningful in Euclidian space such as physical edge betweenness⁴⁸. It is proposed that geometrically constrained brain connectivity emerges as a result of minimized metabolic cost in neurodevelopment, which is explored in computational simulations known as generative models^{49,50}. Additionally, examinations of communication dynamics elucidate the topological mechanisms that scaffold neuronal signaling across networks, providing a plausible bridge between structure and functional states⁵¹. Network control paradigms aim to determine which structural components drive system functionality through perturbative analysis^{52,53}, while other approaches model neuron spiking activity to reveal underlying topologies that could plausibly manifest emergent behavior⁵⁴. Finally, resilience analysis may prove useful in quantifying the extent to which a network can withstand deterioration due to pathology associated with diseases such as schizophrenia or traumatic insult⁵⁵. The majority of these network characterizations have centered on macroscale brain systems, however, future application to microscale neurocircuitry is also warranted.”

References

1. Ganmor, E., Segev, R. & Schneidman, E. Sparse low-order interaction network underlies a highly correlated and learnable neural population code. *Proc Natl Acad Sci U S A* **108**, (2011).
2. Giusti, C., Ghrist, R. & Bassett, D. S. Two's company, three (or more) is a simplex: Algebraic-topological tools for understanding higher-order structure in neural data. *Journal of Computational Neuroscience* vol. 41 Preprint at <https://doi.org/10.1007/s10827-016-0608-6> (2016).
3. Battiston, F. *et al.* Networks beyond pairwise interactions: Structure and dynamics. *Physics Reports* vol. 874 Preprint at <https://doi.org/10.1016/j.physrep.2020.05.004> (2020).
4. Otter, N., Porter, M. A., Tillmann, U., Grindrod, P. & Harrington, H. A. A roadmap for the computation of persistent homology. *EPJ Data Sci* **6**, (2017).
5. Sizemore, A. E., Phillips-Cremins, J. E., Ghrist, R. & Bassett, D. S. The importance of the whole: Topological data analysis for the network neuroscientist. *Network Neuroscience* **3**, (2019).
6. Sizemore, A. E. *et al.* Cliques and cavities in the human connectome. *J Comput Neurosci* **44**, (2018).
7. Spreemann, G., Dunn, B., Botnan, M. B. & Baas, N. A. Using persistent homology to reveal hidden information in neural data. (2015).
8. Lee, H., Kang, H., Chung, M. K., Kim, B. N. & Lee, D. S. Persistent brain network homology from the perspective of dendrogram. *IEEE Trans Med Imaging* **31**, (2012).
9. Berge, C. *Graphs and Hypergraphs*. Elsevier Science Ltd. (1985).
10. Yang, J. *et al.* Constructing high-order functional networks based on hypergraph for diagnosis of autism spectrum disorders. *Front Neurosci* **17**, (2023).
11. Wu, Z. *et al.* A Comprehensive Survey on Graph Neural Networks. *IEEE Trans Neural Netw Learn Syst* **32**, (2021).
12. Bronstein, M. M., Bruna, J., Cohen, T. & Velicković, P. Geometric Deep Learning: Grids, Groups, Graphs, Geodesics, and Gauges. (2021).
13. Bronstein, M. M., Bruna, J., Lecun, Y., Szlam, A. & Vandergheynst, P. Geometric Deep Learning: Going beyond Euclidean data. *IEEE Signal Processing Magazine* vol. 34 Preprint at <https://doi.org/10.1109/MSP.2017.2693418> (2017).
14. Dan, T. *et al.* Uncovering shape signatures of resting-state functional connectivity by geometric deep learning on Riemannian manifold. *Hum Brain Mapp* **43**, (2022).
15. Pun, C. S., Lee, S. X. & Xia, K. Persistent-homology-based machine learning: a survey and a comparative study. *Artif Intell Rev* **55**, (2022).
16. Zia, A. *et al.* Topological Deep Learning: A Review of an Emerging Paradigm. (2023).
17. Hajij, M. *et al.* Topological Deep Learning: Going Beyond Graph Data.
18. Zhao, R., Wang, H., Zhang, C. & Cai, W. PointNeuron: 3D Neuron Reconstruction via Geometry and Topology Learning of Point Clouds. in *Proceedings - 2023 IEEE Winter Conference on Applications of Computer Vision, WACV 2023* (2023). doi:10.1109/WACV56688.2023.00574.
19. Ossinger, A. *et al.* A rapid and accurate method to quantify neurite outgrowth from cell and tissue cultures: Two image analytic approaches using adaptive thresholds or machine learning. *J Neurosci Methods* **331**, (2020).

20. Schmidbauer, D. *et al.* ExplantAnalyzer: An advanced automated neurite outgrowth analysis evaluated by means of organotypic auditory neuron explant cultures. *J Neurosci Methods* **363**, (2021).
21. Broser, P. J. *et al.* Automated axon length quantification for populations of labelled neurons. *J Neurosci Methods* **169**, (2008).
22. Grüning, P. *et al.* Robust and Markerfree in vitro Axon Segmentation with CNNs. in *Lecture Notes of the Institute for Computer Sciences, Social-Informatics and Telecommunications Engineering, LNICST* vol. 362 LNICST (2021).
23. Kandel, M. E. *et al.* Multiscale Assay of Unlabeled Neurite Dynamics Using Phase Imaging with Computational Specificity. *ACS Sens* **6**, (2021).
24. Liu, Z., Cootes, T. & Ballestrem, C. An End to End System for Measuring Axon Growth. in *Lecture Notes in Computer Science (including subseries Lecture Notes in Artificial Intelligence and Lecture Notes in Bioinformatics)* vol. 12436 LNCS (2020).
25. Palumbo, A. *et al.* Deep learning to decipher the progression and morphology of axonal degeneration. *Cells* **10**, (2021).
26. Mello, G. B. M. e *et al.* DeepTEGINN: Deep Learning Based Tools to Extract Graphs from Images of Neural Networks. (2019).
27. Kan, A. Machine learning applications in cell image analysis. *Immunology and Cell Biology* vol. 95 Preprint at <https://doi.org/10.1038/icb.2017.16> (2017).
28. Shepherd, G. M. G., Stepanyants, A., Bureau, I., Chklovskii, D. & Svoboda, K. Geometric and functional organization of cortical circuits. *Nat Neurosci* **8**, (2005).
29. Ho, S. Y. *et al.* NeurphologyJ: An automatic neuronal morphology quantification method and its application in pharmacological discovery. *BMC Bioinformatics* **12**, (2011).
30. Haghighi, F. *et al.* Self-supervised Learning for Segmentation and Quantification of Dopamine Neurons in $\text{\text{Parkinson's Disease}}$. (2023).
31. Mencattini, A. *et al.* NeuroTES. Monitoring neurite changes through transfer entropy and semantic segmentation in bright-field time-lapse microscopy. *Patterns* **2**, (2021).
32. Moreno Mello, G. B. *et al.* Method to Obtain Neuromorphic Reservoir Networks from Images of in Vitro Cortical Networks. in *2019 IEEE Symposium Series on Computational Intelligence, SSCI 2019* (2019). doi:10.1109/SSCI44817.2019.9002741.
33. Manubens-Gil, L. *et al.* BigNeuron: a resource to benchmark and predict performance of algorithms for automated tracing of neurons in light microscopy datasets. *Nat Methods* **20**, (2023).
34. Brown, K. M. *et al.* The DIADEM data sets: Representative light microscopy images of neuronal morphology to advance automation of digital reconstructions. *Neuroinformatics* vol. 9 Preprint at <https://doi.org/10.1007/s12021-010-9095-5> (2011).
35. de Santos-Sierra, D. *et al.* Graph-based unsupervised segmentation algorithm for cultured neuronal networks' structure characterization and modeling. *Cytometry Part A* **87**, (2015).
36. Hardelauf, H. *et al.* Micropatterning neuronal networks. *Analyst* **139**, (2013).
37. de Souza, E. M., Costa, E. T. & Castellano, G. Investigation of anisotropic fishing line-based phantom as tool in quality control of diffusion tensor imaging. *Radiol Phys Technol* (2019) doi:10.1007/s12194-019-00507-9.
38. Lee, J. H. A study on the characteristics of materials for ex vivo phantom of diffusion tensor images. *Journal of the Korean Physical Society* **82**, (2023).

39. Perrin, M. *et al.* Validation of q-ball imaging with a diffusion fibre-crossing phantom on a clinical scanner. *Philosophical Transactions of the Royal Society B: Biological Sciences* **360**, (2005).
40. Cheng, H. *et al.* Characteristics and variability of structural networks derived from diffusion tensor imaging. *Neuroimage* **61**, (2012).
41. Smith, R. E., Tournier, J. D., Calamante, F. & Connelly, A. SIFT: Spherical-deconvolution informed filtering of tractograms. *Neuroimage* **67**, (2013).
42. Smith, R. E., Tournier, J. D., Calamante, F. & Connelly, A. SIFT2: Enabling dense quantitative assessment of brain white matter connectivity using streamlines tractography. *Neuroimage* **119**, (2015).
43. Daducci, A., Dal Palù, A., Lemkaddem, A. & Thiran, J. P. COMMIT: Convex optimization modeling for microstructure informed tractography. *IEEE Trans Med Imaging* **34**, (2015).
44. Yang, R., Sala, F. & Bogdan, P. Hidden network generating rules from partially observed complex networks. *Commun Phys* **4**, (2021).
45. Xiao, X., Chen, H. & Bogdan, P. Deciphering the generating rules and functionalities of complex networks. *Sci Rep* **11**, (2021).
46. Rendón de la Torre, S., Kalda, J., Kitt, R. & Engelbrecht, J. Fractal and multifractal analysis of complex networks: Estonian network of payments. *European Physical Journal B* **90**, (2017).
47. Yin, C. *et al.* Network science characteristics of brain-derived neuronal cultures deciphered from quantitative phase imaging data. *Sci Rep* **10**, (2020).
48. Song, H. F., Kennedy, H. & Wang, X. J. Spatial embedding of structural similarity in the cerebral cortex. *Proc Natl Acad Sci U S A* **111**, (2014).
49. Akarca, D. *et al.* A generative network model of neurodevelopmental diversity in structural brain organization. *Nat Commun* **12**, (2021).
50. Betzel, R. F. & Bassett, D. S. Generative models for network neuroscience: Prospects and promise. *Journal of the Royal Society Interface* vol. 14 Preprint at <https://doi.org/10.1098/rsif.2017.0623> (2017).
51. Seguin, C., Sporns, O. & Zalesky, A. Brain network communication: concepts, models and applications. *Nature Reviews Neuroscience* vol. 24 Preprint at <https://doi.org/10.1038/s41583-023-00718-5> (2023).
52. Yan, G. *et al.* Network control principles predict neuron function in the *Caenorhabditis elegans* connectome. *Nature* **550**, (2017).
53. Reed, E. A., Ramos, G., Bogdan, P. & Pequito, S. The role of long-term power-law memory in controlling large-scale dynamical networks. *Sci Rep* **13**, 19502 (2023).
54. Gupta, G., Rhodes, J., Kiani, R. & Bogdan, P. Neuron particles capture network topology and behavior from single units. *bioRxiv* (2021).
55. Lo, C. Y. Z. *et al.* Randomization and resilience of brain functional networks as systems-level endophenotypes of schizophrenia. *Proc Natl Acad Sci U S A* **112**, (2015).

REVIEWERS' COMMENTS:

Reviewer #2 (Remarks to the Author):

I very much enjoyed reading the revision of this very timely article. There is no article reviewing this important topic of analyzing microscopically networks of neurons. Regarding the geometric deep learning and simplicial complexes as one of the reviewers mentions, it is true that there is a hot area of research and the authors should also consider very recent machine learning developments like "Efficient representation learning for higher-order data with simplicial complexes." In Learning on Graphs Conference, pp. 13-1. PMLR, 2022. Of note, under the "generative models^{49,50}" the authors can also include the weighted multifractal graph generator from already cited papers like . Hidden network generating rules from partially observed complex networks. Commun Phys 4, (2021). or "Controlling the multifractal generating measures of complex networks." Scientific reports 10, no. 1 (2020): 5541. These generative models can retrieve network models like the Erdos-Renyi or Barabasi Albert or Strogatz-Watts as particular cases but also allow to analytically compute the clustering coefficient distribution or the multifractal properties. Overall, this is a very strong review paper and I strongly recommend publication. It is timely, guiding the community and most importantly inspiring new directions at least for my research group.